

# Morphometric and taxonomic approach to describe *Heterospio variabilis* (Annelida, Longosomatidae), a new species with three size-dependent morphotypes, from the Gulf of California, Eastern Pacific

Pablo Hernández-Alcántara[1] and  Vivianne Solis-Weiss[2]

[1] Departmento de Ecología Acuática y Biodiversidad, Instituto de Ciencias del Mar y Limnología, Universidad Nacional Autonoma de México (UNAM), Mexico City, Tlalpan, D.F., Mexico

[2] Departmento de Sistemas Arrecifales, Instituto de Ciencias del Mar y Limnología, Universidad Nacional Autónoma de México, Puerto Morelos, Quintana Roo, Mexico

Corresponding author
Vivianne Solis-Weiss,
solisw@cmarl.unam.mx

## ABSTRACT

The Longosomatidae, a poorly known polychaete family, includes only 23 recognized species; in this study, based on morphometric and taxonomic analyses, we describe a new species with three morphotypes: *Heterospio variabilis* from the Gulf of California, Mexico. The specimens examined exhibit large morphological variations but were clearly separated from close species due to a unique combination of morphological characters: chaetiger 9 as the first elongated chaetiger, four to eight branchial pairs; chaetae from chaetiger 10 forming rings in two rows, posterior row with thin and robust capillaries, anterior row with subuluncini, aristate spines, acicular spines and thick acicular spines. With the discriminant analysis, carried out on 11 morphometric characters, the presence of three morphological groups were recognized (Wilks' lambda= 0.093, $p = 0.0001$). However, the variables selected to discriminate the specimens (partial Wilks' lambda > 0.57) were correlated to their size: number of branchiae, body width, prostomium width, rate length CH9/CH1-CH8, length CH1-CH8 and length CH9 ($r > 0.5$). So, we concluded that they belong to a single species with three morphotypes: morpho A with eight branchial pairs, morpho B with 5–6–7 pairs and morpho C with 4 pairs. No correlations between the distribution of the distinct morphotypes along the eastern gulf shelf and the environmental conditions where they settle were detected.

## INTRODUCTION

At present, the polychaetes comprise nearly 11,500 accepted nominal species (*Pamungkas et al., 2019*; *Read, 2019*), and are one of the most abundant and diverse groups of invertebrates in soft marine bottoms worldwide (*Hutchings, 1998*; *Brooks et al., 2006*; *Pamungkas et al., 2019*; *Pamungkas, Glasby & Costello, 2021*). However, according to *Read (2019)*, it could be

that as much as 60–65% of the species remain undescribed, thus limiting our knowledge of their role on the ecology and evolutionary processes operating in benthic ecosystems (*Martin et al., 2022*). This is the case of the Longosomatidae (*Hartman, 1944*), a marine benthic family poorly known worldwide, and particularly in the Gulf of California, where only one valid species, *H. catalinensis* (*Hartman, 1944*) collected at 1,550–1,590 m depth by *Mendez (2006)* and *Mendez (2007)* and an unnamed species (*Heterospio* sp. 1) has been reported (*Hernández-Alcántara & Solís-Weiss, 2005*).

The family Longosomatidae includes small polychaetes with few segments, characterized by a short anterior region (usually named thorax) with eight to nine chaetigerous segments bearing one to eight pairs of elongated branchial filaments; a middle-body region having very long segments with almost complete rings of chaetae around the body, which include thickened aristate, subuluncini-like capillaries, subuluncini and/or spines; and a posterior bulbous region with three to five short segments. The posterior region has not been recorded for several species, because they are usually lost during the sampling process (*Borowski, 1994*; *Parapar, Aguirrezabalaga & Moreira, 2014*; *Hernández-Alcántara & Solís-Weiss, 2021*; *Blake & Maciolek, 2023*).

The first species, *Heterospio longissima*, was described from an incomplete specimen collected in 1869 during the "Porcupine" expedition in deep-sea of Irish waters, and its describer, *Ehlers (1875)*, placed it in the family Spionidae (*Blake & Maciolek, 2023*). Many years later, *Hartman (1944)* described a new genus and species, *Longosoma catalinensis*, from southern California, placing it in a new family, Longosomidae. Later, the same *Hartman (1963)* recognized that Longosomidae should be referred to a new family, Heterospionidae, because the only genus, *Longosoma Hartman, 1944*, should be synonymized with *Heterospio* Ehlers, 1985. However, this priority reversal of a family name is incorrect (*Read & Fauchald, 2024*) and *Borowski (1994)*, describing a new species and summarizing the knowledge on the taxonomy and biology of this family, pointed out that Longosomidae (*Hartman, 1944*), as the older name, had priority on Heterospionidae (*Hartman, 1963*), and erected the correct name of Longosomatidae, amending the original spelling of the name (*Rouse, Pleijel & Tilic, 2022*). Nevertheless, *Borowski (1994)* appears to state *Heterospio longissima* as the "type species", though a family does not have a type species and *Longosoma* should remain as the type genus, regardless of its synonymy with *Heterospio* (*Read & Fauchald, 2024*).

Recently, *Blake & Maciolek (2019)* reviewed the history and biology of longosomatids and *Blake & Maciolek (2023)* carried out an excellent and detailed study of specimens from various biological surveys from the North Atlantic, Gulf of Mexico, Caribbean Sea, off Brazil, off California, the Indian Ocean, New Zealand, Australia, and South China. In the latter publication, they described 13 new species and presented new descriptions and records of *H. catalinensis* (*Hartman, 1944*), *H. indica Parapar et al., 2016* and *H. peruana Borowski, 1994*. They also examined the original material which *Hartman (1965)* described as *H. longissima* from the Atlantic Ocean. They found significant differences between the published description and the examined specimens, such as the number of branchiae, the prostomium shape, the origin of the chaetal fascicles or the chaetal types, concluding that

the specimens belonged to two separate new species: *H. hartmanae* from abyssal depths off New England to Bermuda and *H. guiana* from the upper slope depths off Surinam.

In the Mexican Pacific, only *H. catalinensis* (*Mendez, 2006*; *Mendez, 2007*) from deep-sea and *Heterospio* sp. 1 (*Hernández-Alcántara & Solís-Weiss, 2005*) from the Gulf of California had been recorded, but when the individuals of *Heterospio* sp. 1, together with other specimens collected in three oceanographic expeditions, were examined by us with more detail, we found that they all belong to a new species. Therefore, the aim of the present study was to describe this new species, supported with scanning electron microscopy (SEM) photographs and to confirm it as a new taxon. We also assess its intraspecific variability with statistical multivariate analysis. This study is based on 11 selected quantitative morphometric characteristics, focusing on the likely relation between the morphological variability and the body size. The inclusion of quantitative characters to carry out morphometric analyses has been seldom used in studies aimed at describing new polychaete species. Nevertheless, we observed that the addition of quantitative features provided important additional information to support the discrimination between polychaete taxa, mainly when the examined specimens exhibit morphological variability. For example, we used multivariate morphometric analysis to separate closely related species and erect the new orbinid *Leitoscoloplos multipapillatus* (*Hernández-Alcántara & Solís-Weiss, 2014*); also, *Hernández-Alcántara, Mercado-Santiago & Solís-Weiss (2017)* analyzed the morphometric variability of the onuphid *Paradiopatra multibranchiata* to determine the main characters supporting its description as a new species; *Martin et al. (2022)* described the new terebellid *Loimia davidi*, examining the taxonomic implications of size-dependent morphological characters and its differences with close species using hypervolume analysis, and summarizing the morphological information of all known species of *Loimia*; *Molina-Acevedo, Fernández-Rodríguez & Idris (2022)* carried out a morphometric approach to separate the eunicids *Paucibranchia belli*, *P. disjuncta* and *P. carrerai*, evaluating the importance of 23 morphological features and their correlation with the individuals' size to discriminate species within the genus. Additionally, the relationships between the environmental variables and the distribution of the new species along the Gulf of California shelf were also examined to improve the knowledge about the poorly known ecology of the longosomatids.

## MATERIALS & METHODS

### Samples collection and morphological examination

The biological material was collected during three oceanographic expeditions carried out in the Gulf of California (Mazatlán Bay in May 1980; Cortes 2 in March 1985; Cortes 3 in August 1985) (Fig. 1, Table 1). All of these expeditions were conducted on board the O/V "El Puma" of the Instituto de Ciencias del Mar y Limnología (ICML), Universidad Nacional Autónoma de México (UNAM). The stations were georeferenced on-board with a Global Positioning System (GPS) and depth was measured with a Kongsberf echosounder. In the Mazatlán Bay expedition, the material was collected with a Van Veen grab (0.063 m$^2$), the temperature was measured with a thermometer (±0.05 °C) and the salinity with a conductivity sensor Plessey (Model 6230). During the Cortes 2 and 3 expeditions,

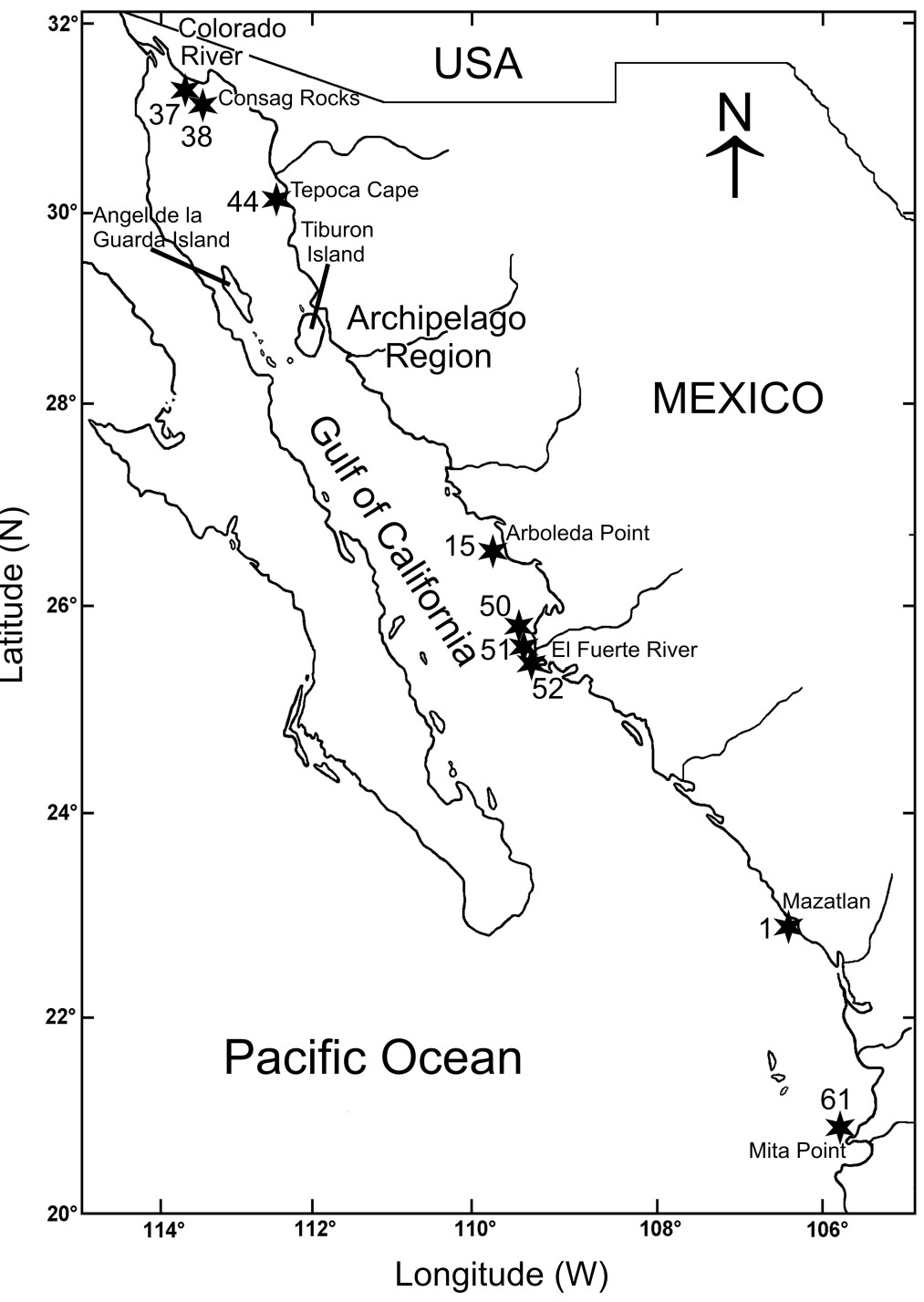

**Figure 1** Gulf of California including the sampling stations where *Heterospio variabilis* sp. nov. was collected.

**Table 1  Location, environmental data and number of individuals of *Heterospio viariabilis* sp. nov. by sampling station in the Gulf of California.**

| Station | Date | Latitude (N) | Longitude (W) | Depth (m) | Salinity (psu) | Temperature (°C) | Dissolved oxygen (ml/L) | Organic matter (%) | Sand (%) | Sediment type | Number of specimens |
|---|---|---|---|---|---|---|---|---|---|---|---|
| 37-Cortes 2 | 1985-03-16 | 31°16.1′ | 114°21.7′ | 30.3 | 35.51 | 16.0 | 5.40 | 2.4 | 85 | Fine sand | 30 |
| 38-Cortes 2 | 1985-03-16 | 31°08.3′ | 114°13.3′ | 71.9 | 35.45 | 14.5 | 3.17 | — | — | — | 2 |
| 52-Cortes 2 | 1985-03-20 | 25°39.9′ | 109°28.6′ | 28.6 | 35.19 | 16.8 | 5.40 | 3.6 | 58 | Fine sand | 2 |
| 51-Cortes 2 | 1985-03-20 | 25°42.1′ | 109°30.6′ | 49.5 | 35.15 | 14.8 | 1.80 | 7.2 | 58 | Fine sand | 1 |
| 50-Cortes 2 | 1985-03-20 | 25°46.8′ | 109°35.4′ | 97.0 | 34.99 | 13.2 | 1.47 | 5.7 | 62 | Fine sand | 10 |
| 61-Cortes 2 | 1985-03-23 | 20°53.9′ | 105°27.5′ | 50.4 | 34.92 | 16.8 | 1.03 | 5.5 | 94 | Fine sand | 1 |
| 37-Cortes 3 | 1985-08-04 | 31°19.8′ | 114°23.2′ | 21.5 | 36.06 | 29.6 | 4.26 | 5.00 | 91 | Very fine sand | 3 |
| 44-Cortes 3 | 1985-08-05 | 30°00.5′ | 112°59.5′ | 106.0 | 35.63 | 19.4 | 2.56 | 8.40 | 52 | Very fine sand | 2 |
| 15-Cortes 3 | 31-07-1985 | 26°53.2′ | 110°05.9′ | 39.0 | 34.80 | 28.1 | 3.83 | 6.10 | 81 | Fine sand | 2 |
| 52-Cortes 3 | 1985-08-08 | 25°43.6′ | 109°29.3′ | 22.1 | 34.20 | 30.0 | 4.34 | 5.30 | 83 | Very fine sand | 1 |
| 50-Cortes 3 | 1985-08-08 | 25°49.5′ | 109°37.9′ | 80.0 | 35.22 | 17.6 | 2.22 | 3.80 | 49 | Very fine sand | 1 |
| C8-7 | 1980-01-25 | 23°14.2′ | 106°26.8′ | 7.0 | 34.78 | 23.4 | — | — | — | — | 1 |

the samples were collected with a Smith-McIntyre grab (0.1 m$^2$) and, at each station, temperature and salinity were measured with a Niels Brown CTD, and the dissolved oxygen with the Winkler method (*Strickland & Parsons, 1977*). Additional sedimentary samples were taken to quantify the organic matter content by the *Walkley & Black (1934)* acid digestion method, and the sediment textural characteristics following the sieving method of *Folk (1980)* (Table 1).

The biological samples were washed on-board through a 0.5 mm mesh and fixed with 4% formalin in seawater solution. Later, in the laboratory, the material was washed again with fresh water to eliminate the formalin and the specimens were sorted under a stereoscope and preserved in 70% ethanol. The observations, drawings and measurements of the specimens and their morphological characteristics were made with stereoscope and compound microscopes.

The methyl green staining pattern was determined by immersing the specimens for two minutes in a saturated solution of methyl green in 70% ethanol (*Warren, Hutchings & Doyle, 1994*). Scanning electron microscopy (SEM) observations and micrographs were made using a JEOL JSM6360L microscope at the ICML, UNAM. Specimens were dehydrated *via* graded ethanol series, dried with liquid-CO$_2$ at critical point and coated with gold. The holotype, paratypes and additional material examined were catalogued and deposited in the Colección Nacional de Anélidos Poliquetos, ICML-UNAM. Additional paratypes were also deposited in the Natural History Museum of Los Angeles County (NHMLA).

The species description was based on the holotype, but the morphological variability associated to paratypes was also included in parentheses. A total of 56 specimens of the new species were examined and their occurrence along the Gulf of California is shown in the Material examined section. To standardize the description of the longosomatid species, in general we followed the formats suggested by *Parapar, Aguirrezabalaga & Moreira (2014)*, and *Parapar et al. (2016)* to describe the morphology of the new species but the

terminology is based on *Rouse, Pleijel & Tilic (2022)* and *Blake & Maciolek (2019)*, *Blake & Maciolek (2023)*. The number of segments in the anterior body ("thoracic region") and the relative length of the following elongated segments ("mid-body region") are two significant characteristics to separate the longosomatid species. However, in this family, the intersegmental channels are usually not so evident and, therefore, the separation of segments and the transition between the anterior the middle body region are difficult to distinguish. Therefore, also following *Parapar, Aguirrezabalaga & Moreira (2014)*, and *Parapar et al. (2016)*, the limits between segments were established based on the position of the chaetae on the anterior edge of the segments and, therefore, the length of a segment is the distance from the chaetal bundle of one chaetiger to the chaetal bundle of the next.

Due to the fact that new evidence has shown that the Longosomatidae are more closely related to cirratulids than spionids (*Rouse, Pleijel & Tilic, 2022*; *Blake & Maciolek, 2023*), *Blake & Maciolek (2023)* suggested the use of "dorsal tentacles" to name the grooved feeding structures in this family. However, until their phylogenetic relationships can be fully clarified, we considered more appropriate to maintain here the term "palps" as has been commonly used in the descriptions of the longosomatid species (*Hartman, 1965*; *Wu & Chen, 1966*; *Borowski, 1994*; *Blake & Maciolek, 2019*, among others).

Despite the small number of described longosomatid species, their types of chaetae are highly variable. Although *Borowski (1994)* and *Parapar, Aguirrezabalaga & Moreira (2014)* discussed some aspects of their chaetal variability, it is necessary to examine in detail their shapes and to review the terminology used, since, as *Borowski (1994)* and *Parapar et al. (2016)* pointed out, some chaetal types could be transitional stages of the same chaeta. Therefore, and due to the wide chaetal variety observed in the new species, we kept the terms currently used to characterize the chaetal types: capillary, stout capillary, subuluncini, aristate spines, acicular spines and hooks (*Borowski, 1994*; *Parapar et al., 2016*).

The electronic version of this article in Portable Document Format (PDF) will represent a published work according to the International Commission on Zoological Nomenclature (ICZN), and hence the new names contained in the electronic version are effectively published under that Code from the electronic edition alone. This published work and the nomenclatural acts it contains have been registered in ZooBank, the online registration system for the ICZN. The ZooBank LSIDs (Life Science Identifiers) can be resolved and the associated information viewed through any standard web browser by appending the LSID to the prefix http://zoobank.org/. The LSID for this publication is: urn:lsid:zoobank.org:pub:F3462F09-2330-42F3-BA76-C2ACDEE10504. The online version of this work is archived and available from the following digital repositories: PeerJ, PubMed Central SCIE and CLOCKSS.

## Anatomical and morphometric comparisons

In order to examine the taxonomy and morphological variability of the new species, 11 characters were measured from the 56 specimens mentioned: number of branchiae (brN); prostomium length (prL) and width (prW); length of chaetiger 1 to chaetiger 8 (ch1-ch8L); width of body (at chaetiger 5 without parapodia) (anW); length of chaetiger 9, the first elongated segment (ch9L); length of chaetiger 10 (ch10L), 11 (ch11L) and 12 (ch12L); the

measure of the relative length of the first elongated chaetiger; the rate between the length of chaetiger 9 and the length of the anterior region (tip of prostomium to end of chaetiger 8) (Rch9/anL). For comparative purposes, and because all the specimens were incomplete and had suffered mechanical damage during the collection and sieving processes, and because 88% of the specimens had at least 12 chaetigers, the total size of individuals was standardized to the length back to the 12th chaetiger, naming it here: "total length" (tL12). The data that support the findings of this research are openly available online through the SEANOE portal at https://doi.org/10.17882/96906 (*Hernández-Alcántara & Solís-Weiss, 2023*).

An estimation of the descriptive statistic parameters (mean, range, standard deviation (SD) and coefficient of variation (CV) was carried out to examine the variability of the characters measured. To determine whether the morphological variability was linked to the body size of the specimens, we examined the correlations between the total length (tL) and each morphological variable with the Pearson correlation coefficient ($r$).

The morphological relations between the examined specimens, based on the 11 measured morphometric variables, were calculated by means of a principal component analysis (PCA). The morphometric distinction between the specimens was made with a Discriminant Analysis using the forward stepwise method; the standard statistic Wilks' lambda (ranging from 1, no discriminatory power, to 0, perfect discriminatory power) was used to assess the characters that most significantly contributed to differentiate the specimens' groups. The partial Wilks' lambda index was used to evaluate the individual contribution of each measured character to discriminate the groups. The graphical representation for distinction among the specimen groups was performed with a canonical analysis (STATISTICA 7.0; StatSoft, Inc., Tulsa, OK, USA). The relationships between the environmental conditions and the distribution of the morphological groups in the Gulf of California shelf were estimated with PCA. All morphometric analyses were carried out with the STATISTICA 7.0 software (StatSoft, Inc., Tulsa, OK, USA) for Windows.

## RESULTS

### Systematics

Family Longosomatidae *Hartman, 1944*

Genus *Heterospio Ehlers, 1874*

*Heterospio variabilis* **sp. nov.**

LSID:  urn:lsid:zoobank.org:act:C8DF52A4-B1F1-4F9B-83C4-5A410414A9E7

(Figs. 2A–2J, 3A–3K, 4A–4H, 5A–5K, 6A–6L, 7A–7L)

*Heterospio* sp. 1.–*Hernández-Alcántara & Solís-Weiss, 2005*: 277.

**Material examined**. *Type locality*. MEXICO ●Gulf of California, North Consag Rocks; 31°16.1′N, 114°21.7′W; 30.3 m. *Holotype*: MEXICO ●1 spec.; Gulf of California, North Consag Rocks, Sta. 37 Cortes 2; 31°16.1′N, 114°21.7′W; 30.3 m; 1985 Mar. 16; P. Hernández-Alcántara leg.; fine sand sediment; CNAP-ICML: POH–13–001. *Paratypes:* MEXICO ●5 specs.; Gulf of California; same collection data as for holotype; CNAP-ICML: POP–13–001 (one specimen coated with gold for SEM studies; CNAP-ICML: MEB-POP–13–001) ●2 specs.; Gulf of California, El Fuerte River, Sta. 52 Cortes 2; 25°39.9′N,

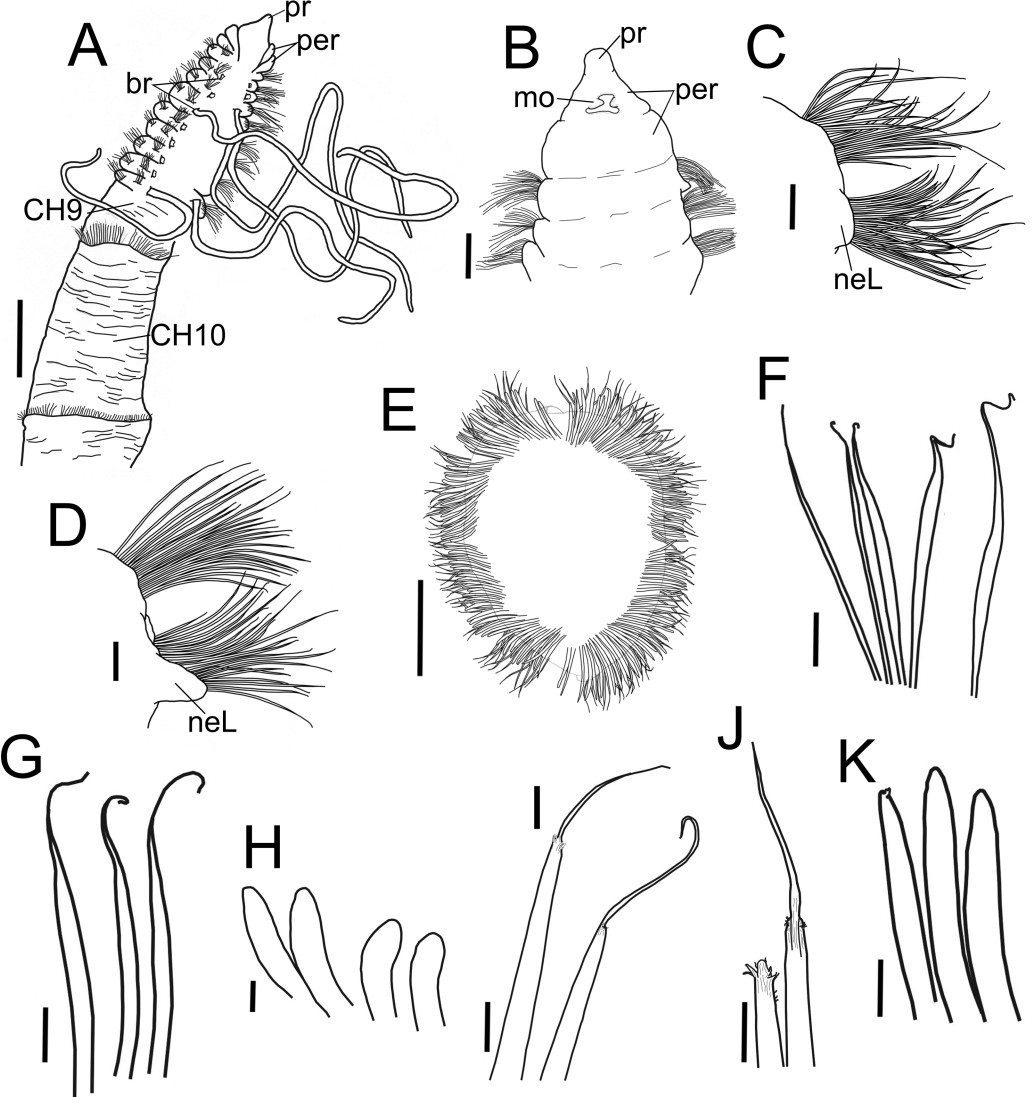

**Figure 2** *Heterospio variabilis* **sp nov.** (A) Anterior and middle body, dorsal view. (B) Anterior end, ventral view. (C) Chaetiger 1. (D) Chaetiger 9. (E) Chaetiger 12, cross section. (F) Thin capillaries and robust capillaries flattened in middle half. (G) Subuluncini. (H) Thick acicular spines. (I) Aristate spines. (J) Detail of distal end of aristate spines, one with the aristate end missing. (K) Acicular spines. A, holotype (CNAP-ICML: POH–13–001); B-K, paratype (CNAP-ICML: POP–13–001). Scale bars: A, E = 500 μm; B, C, D = 100 μm; F, G, K = 20 μm; H, I, J = 10 μm. Abbreviations: br, branchia; mo, mouth; neL, neuropodial lamella; per, peristomium; pr, prostomium.

109°28.6′W; 28.6 m; 1985 Mar. 20; P. Hernández-Alcántara leg.; fine sand sediment; CNAP-ICML: POP–13–002 ●4 specs.; Gulf of California, North Consag Rocks, Sta. 37 Cortes 2; 31°16.1′N, 114°21.7′W; 30.3 m; 1985 Mar. 16; P. Hernández-Alcántara leg.; fine sand sediment; LACM-AHF-POLY 14395.

**Additional material**. MEXICO ●20 specs.; Gulf of California, North Consag Rocks, Sta. 37 Cortes 2; 31°16.1′N, 114°21.7′W; 30.3 m; 1985 Mar. 16; P. Hernández-Alcántara

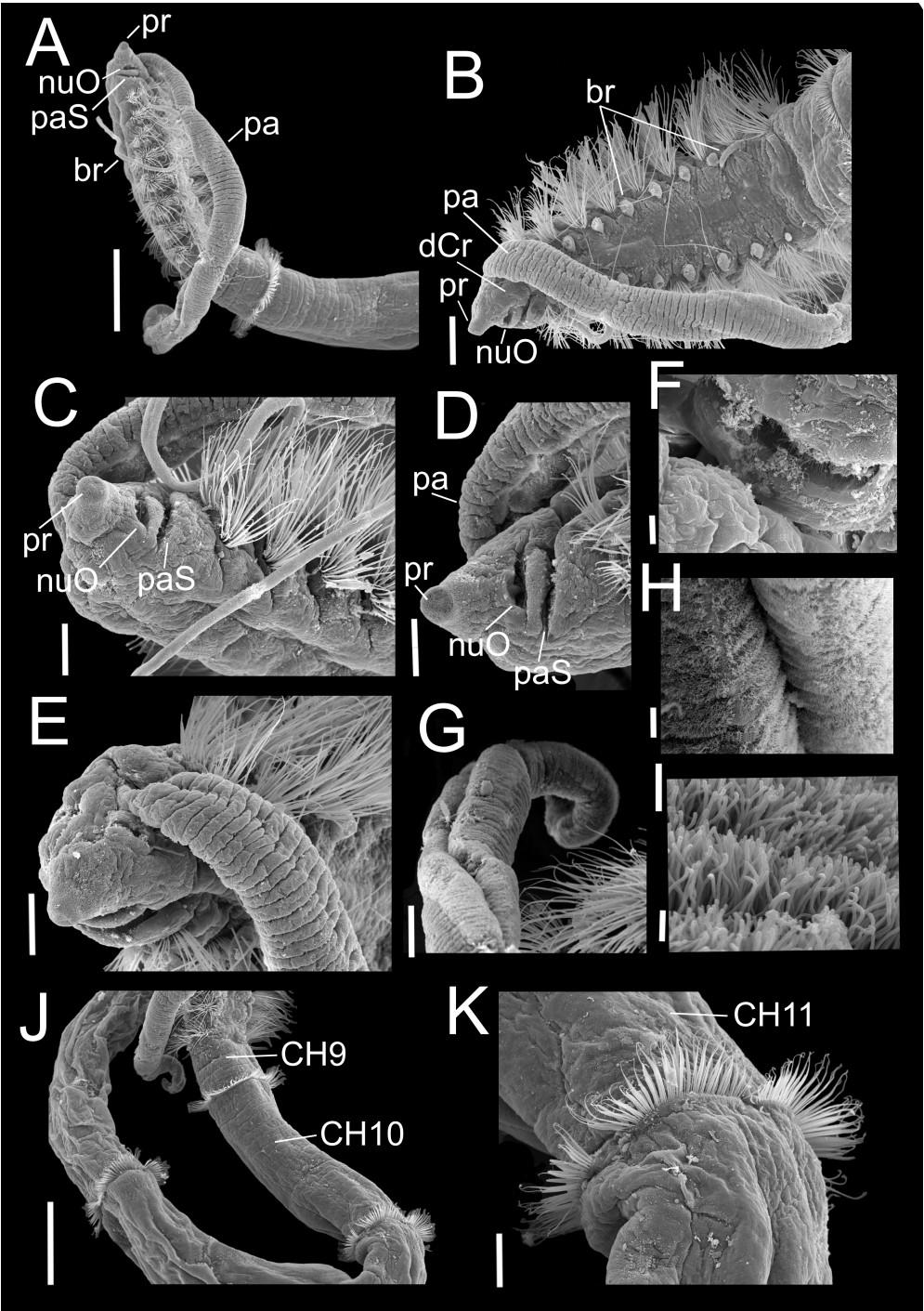

**Figure 3** *Heterospio variabilis* **sp. nov.** Paratype (CNAP-ICML: MEB-POP–13–001). (A) Anterior and middle body, lateral view. (B) Anterior end, dorsal view. (C) Anterior end, ventral view. (D) Detail of prostomium, nuchal organ and palp scar, lateral view. (E) Detail of prostomium and palp. (F) Detail of nuchal organ opening. (G) Palp. (H) Detail of palp, middle region. (I) Detail of cilia on the middle palp. (J) Chaetigers 9–12. (K) Chaetiger 11. Scale bars: A, J = 500 μm; B = 200 μm; C, E, G, K = 100 μm; D = 100 μm; F, H = 10 μm; I = 1 μm. Abbreviations: br, branchia; CH9, 10, 11, chaetigers 9, 10, 11; dCr, dorsal crest; pa, palp; paS, palp scar; nuO, nuchal organ; pr, prostomium.

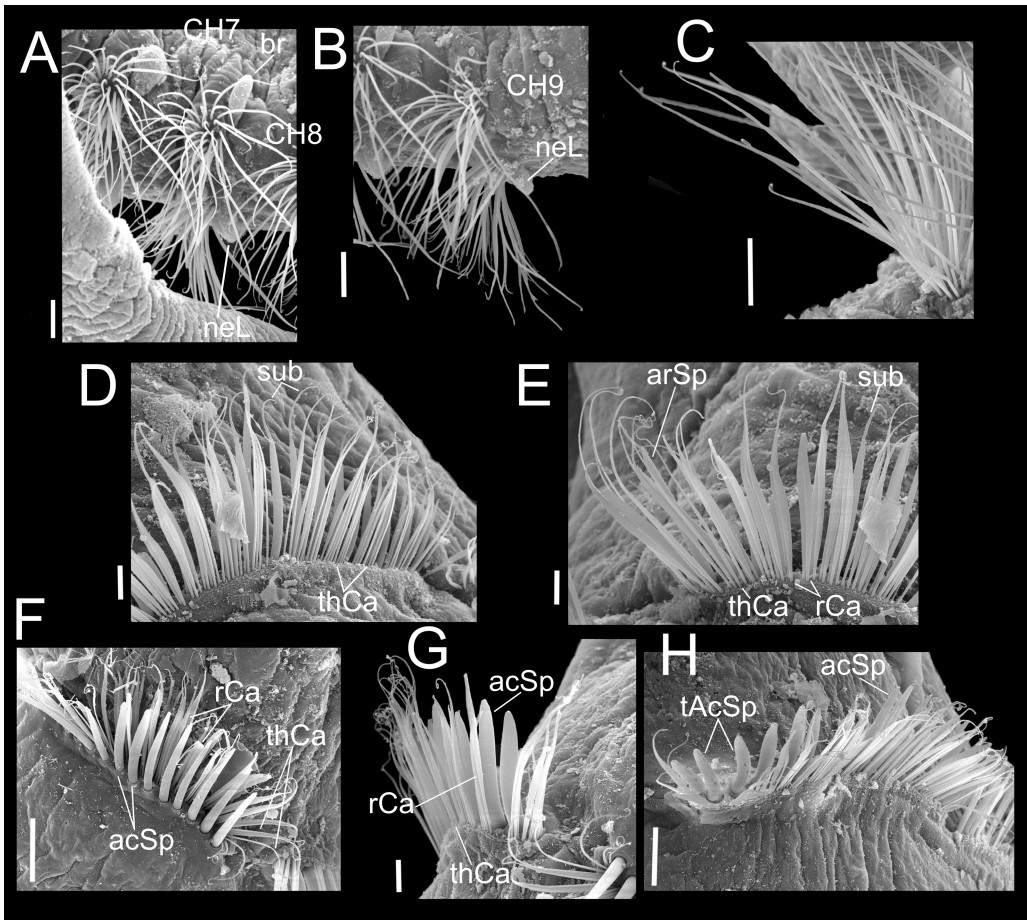

**Figure 4** *Heterospio variabilis* **sp. nov.** Paratype (CNAP-ICML: MEB-POP–13–001). (A) Chaetigers 7–8. (B) Chaetiger 9. (C) Notopodia chaetiger 9. (D) Chaetae of chaetiger 10. (E) Chaetae of chaetigers 11. (F) Chetae of chaetiger 12. (G) Chaetae of chaetiger 12. (H) Chaetae of chaetiger 14. Scale bars: A, B, C, F, H = 50 μm; D, E, G = 20 μm. Abbreviations: acSp, acicular spine; arSp, aristate spine; br, branchia; neL, neuropodial lamella; rCa, robust capillary flattened in distal half; sub, subuluncini; tAcSp, thick acicular spine; thCa, thin capillary.

leg.; fine sand sediment; CNAP-ICML: PO–13–004/2014-GCA-CS ●1 spec.; Gulf of California, El Fuerte River, Sta. 51 Cortes 2; 25°42.1′N, 109°30.6′W; 49.5 m; 1985 Mar. 20; P. Hernández-Alcántara leg.; fine sand sediment; CNAP-ICML: PO–13–004/2015-GCA-CS ●1 spec.; Gulf of California, Mita Point, Sta. 61 Cortes 2; 20°53.9′N, 105°27.5′W; 50.4 m; 1985 Mar. 23; P. Hernández-Alcántara leg.; fine sand sediment; CNAP-ICML: PO–13–004/2016-GCA-CS ●2 specs.; Gulf of California, North Consag Rocks, Sta. 38 Cortes 2; 31°08.3′N, 114°13.3′W; 71.9 m; 1985 Mar16.; P. Hernández-Alcántara leg.; CNAP-ICML: PO–13–004/2017-GCA-CS ●10 specs.; Gulf of California, El Fuerte River, Sta. 50 Cortes 3; 25° 46.8′N, 109°35.4′W; 97 m; 1985 Mar. 20; P. Hernández-Alcántara leg.; fine sand sediment; CNAP-ICML: PO–13–004/2018-GCA-CS (one specimen coated with gold for SEM studies; CNAP-ICML: MEB-PO–13–004/2018-GCA-CS) ●1 spec.; Gulf of California, El Fuerte River, Sta. 52 Cortes 3; 25°43.6′N, 109°29.3′W; 22.1 m; 1985 Aug. 8; P.

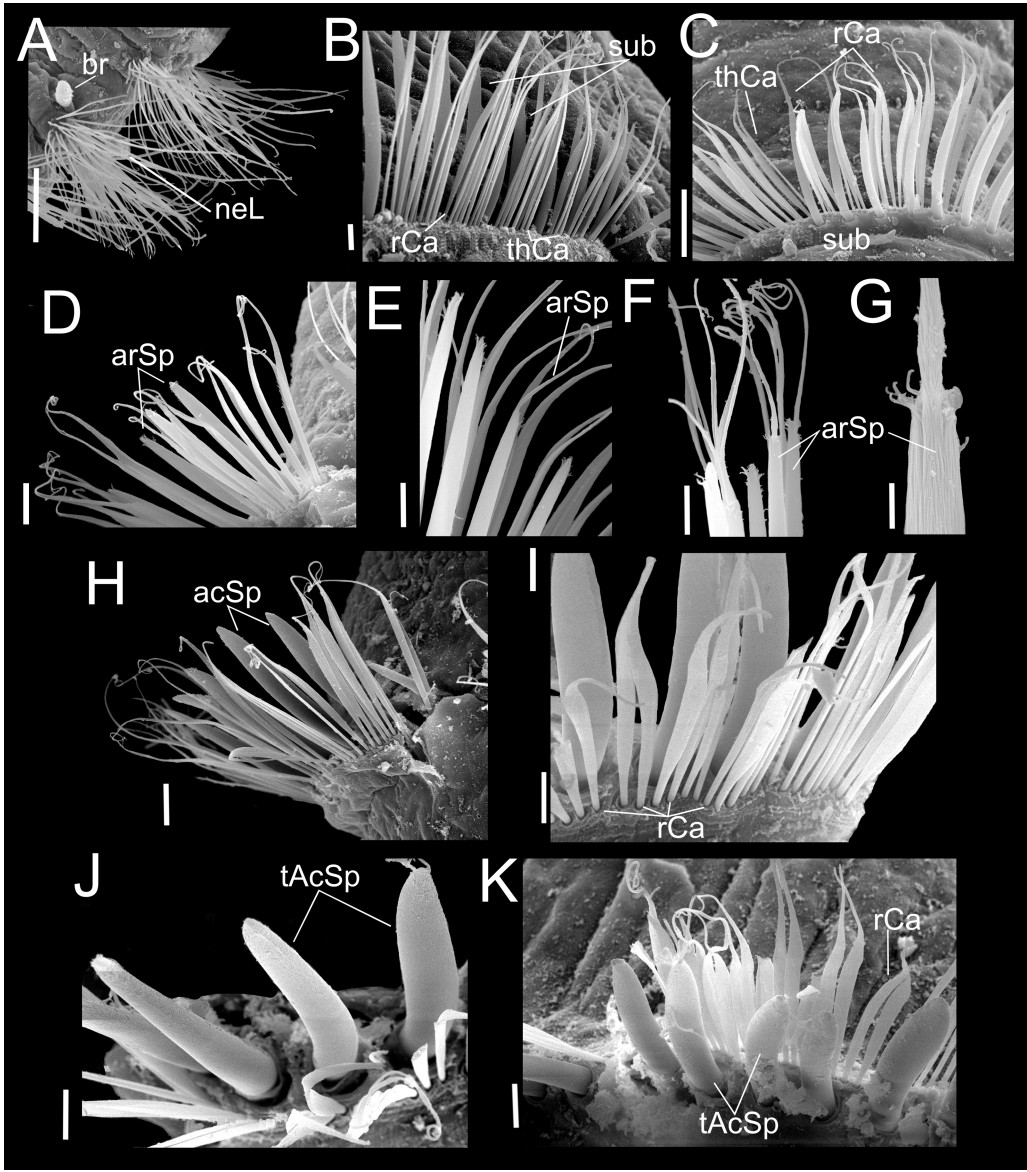

**Figure 5** *Heterospio variabilis* **sp. nov.** Paratype (CNAP-ICML: MEB-POP–13–001). (A) Capillaries of chaetigers 8–9. (B) Subluncini, thin and robust capillary chaetae of chaetiger 10. (C) Subuluncini, thin and robust capillary chaetae of chaetiger 11. (D) Aristate spines without distal appendage of chaetiger 11. (E) Aristate spines of chaetiger 11. (F, G) Detail of distal end of aristate spines. (H) Acicular spine of chaetiger 14. (I) Robust capillaries flattened in the distal half, of chatiger 15. (J) Thick acicular spines of chaetiger 15. (K) Thick acicular spines and robust capillaries of chaetiger 16. Scale bars: A = 100 μm; B, E, F, I, J, K = 10 μm; C, D, H = 20 μm; G = 2 μm. Abbreviations: acSp, acicular spine; arSp, aristate spine; br, branchia; rCa, robust capillary flattened in distal half; sub, subuluncini; tAcSp, thick acicular spine; thCa, thin capillary.

Hernández-Alcántara leg.; very fine sand sediment; CNAP-ICML: PO–13–004/2019-GCA-CS ●2 specs.; Gulf of California, Tepoca Cape, Sta. 44 Cortes 3; 30°00.5′N, 112°59.5′W; 106 m; 1985 Aug. 5; P. Hernández-Alcántara leg.; CNAP-ICML: PO–13–004/2020-GCA-CS ●3

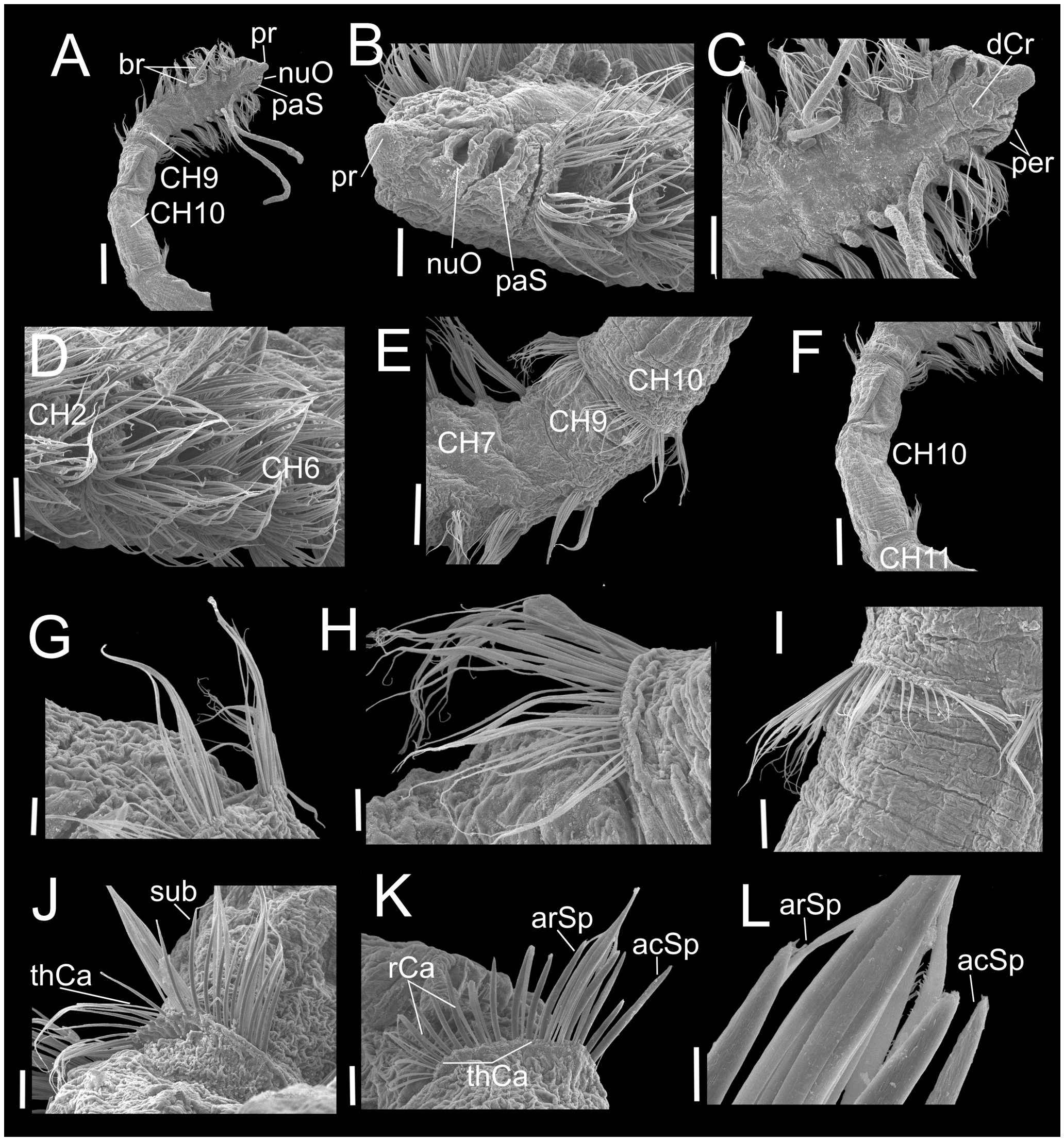

**Figure 6** ***Heterospio variabilis*** **sp. nov., with four branchial pairs (CNAP-ICML: MEB-PO–13–004/2018-GCA-CS).** (A) Anterior and middle region, dorsal view. (B) Anterior region, lateral view. (C) Anterior region, dorsal view. (D) Chaetigers 2–6. (E) Chaetigers 7–10. (F) Chaetiger 10. (G, H) Dorsal and ventral capillaries of chaetiger 10. (I) Chaetiger 11. (J) Subuluncini and thin capillaries of chaetiger 12. (K) Capillaries, acicular and aristate spines of chaetiger 13. (L) Aristate and acicular spines of chaetiger 13. Scale bars: A, F = 200 µm; B, D, I = 50 µm; C, E = 100 µm; G, H, J, K = 20 µm; L = 5 µm. Abbreviations: acSp, acicular spine; arSp, aristate spine; br, branchia; CH2, 6, 7, 9, 10, 11, chaetigers 2, 6, 7, 9, 10, 11; dCr, dorsal crest; pa, palp; paS, palp scar; nuO, nuchal organ; per, peristomium; pr, prostomium; rCa, robust capillary flattened in distal half; sub, subuluncini; thCa, thin capillary.

specs.; Gulf of California, North Consag Rocks, Sta. 37 Cortes 3; 31°19.8′N, 114°23.2′W; 21.5 m; 1985 Aug. 4; P. Hernández-Alcántara leg.; very fine sand sediment; CNAP-ICML: PO–13–004/2021-GCA-CS •2 specs.; Gulf of California, Arboleda Point, Sta. 15 Cortes 3; 26°53.2′N, 110°05.9′W; 39 m; 1985 Jul. 31; P. Hernández-Alcántara leg.; fine sand sediment; CNAP-ICML: PO–13–004/2022-GCA-CS (one specimen coated with gold for

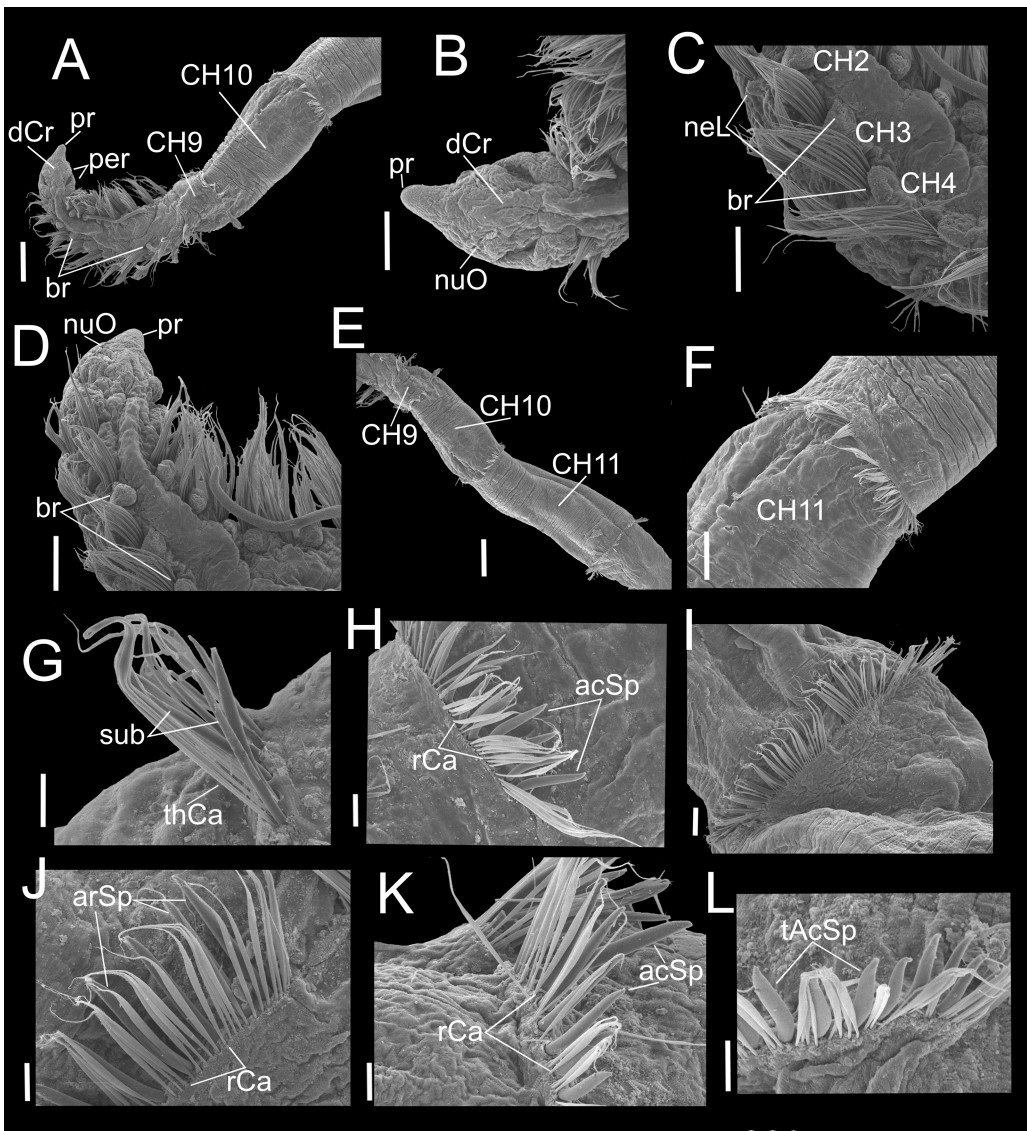

**Figure 7 *Heterospio variabilis* sp. nov. with six branchial pairs (CNAP-ICML: MEB-PO–13–004/2022-GCA-CS).** (A) Anterior and middle region, dorsal view. (B) Prostomium, dorsal view. (C) Chaetigers 2–4. (D) Anterior and branchial region, dorsal view. (E) Chaetigers 9–11. (F) Chaetiger 11. (G) Thin capillaries and subuluncini of chaetiger 11. (H) Robust capillaries flattened in middle half and acicular spines of chaetiger 12. (I) Chaetiger 13. (J) Robust capillaries and aristate spines of chaetiger 13. (K) Robust capillaries and acicular spines of chaetiger 14. (L) Thick acicular spines of chaetiger 14. Scale bars: A, E = 200 µm; B, C, D, F = 100 µm; G, H = 20 µm; I = 50 µm; J, K, L = 20 µm. Abbreviations: acSp, acicular spine; arSp, aristate spine; br, branchia; dCr, dorsal crest; neL, neuropodial lamella; nuO, nuchal organ; per, peristomium; pr, prostomium; rCa, robust capillary flattened in distal half; sub, subuluncini; tAcSp, thick acicular spine; thCa, thin capillary.

SEM studies; CNAP-ICML: MEB-PO–13–004/2022-GCA-CS) ●1 spec.; Gulf of California, El Fuerte River, Sta. 50 Cortes 3; 25°49.5′N, 109°37.9′W; 80 m; 1985 Aug. 8; P. Hernández-Alcántara leg.; very fine sand sediment; CNAP-ICML: PO–13–004/2023-GCA-CS ●1 spec.;
Gulf of California, Mazatlan Bay, Sta. C8–7; 235°14.2′N, 106°26.8′W; 7 m; 1980 Jan. 25; E. Arias-González leg.; CNAP-ICML: PO–13–004/2024-GCA-CS.

**Diagnosis**. Body elongated, threadlike. Anterior region with eight short chaetigers; median region with greatly elongated segments. Chaetiger 9 is the first elongated segment, about twice as long as chaetiger 8. Prostomium conical, anteriorly rounded, continuing as a dorsal crest over peristomium until chaetiger 1; a pair of lateral nuchal organs. Peristomium with two rings interrupted dorsally by dorsal crest; one pair of grooved dorsal palps. Anterior region with 4–8 pairs of long, cirriform branchiae from chaetiger 2, their number related to body size. First nine chaetigers biramous with only simple capillaries, without neuropodial acicular spines. From chaetiger 10 with chaetae forming cinctures, arranged in two rows: anterior row with subuluncini with long distal end, aristate spines, acicular spines, and thick, curved acicular spines; posterior row with thin capillaries and robust capillaries flattened in distal half.

**Description**. Holotype incomplete with 13 chaetigers, 18.6 mm long, 3.7 mm width; 11 paratypes anterior fragments with 12–19 chaetigers, 8.2–24.5 mm long, 2–3.8 mm wide. Body elongated, threadlike (Figs. 2A, 3A), pale in ethanol without special pigmentation. Prostomium conical, anteriorly rounded, slightly flattened dorsoventrally, continuing as a mid-dorsal crest over peristomium until chaetiger 1 (Figs. 3A–3E); eyes absent. Nuchal organs deep grooves, (Fig. 3D) with cilia (Fig. 3F). Peristomium with two rings, interrupted dorsally by crest ridge (Figs. 3B, 3E). One pair of grooved dorsal palps easily deciduous; one paratype (CNAP-ICML: MEB-POP–13–001) with single dorsal palp arising from the right side of peristomium (Figs. 3E, 3G), lined with cilia (Figs. 3H, 3I). Mouth in mid-ventral position at the level of the first peristomial ring, consisting of a simple opening between two large lateral lobes (Fig. 2B). Everted proboscis not observed in any individuals.

Anterior body region slightly flattened dorsoventrally, with eight short chaetigers (CH1–CH8) (Figs. 2A, 3A, 3B), more than twice as wide as long (Figs. 2A, 3B). Chaetiger 9 (CH9) first elongated, longer than wide, about twice as long as CH8 (Figs. 2A, 3B, 3J). Chaetiger 1 (CH1) without branchiae (Figs. 2A, 2C, 3B); with 8 pairs of filiform branchiae from CH2 to CH9, dorsal to notopodia (Figs. 2A, 3B) (7–8 pairs in paratypes, 4–8 pairs in additional material); most branchiae missing, scars difficult to observe. From CH10, segments strongly elongated (Fig. 3A), length progressively increasing towards posterior segments (Fig. 3J): CH10 3.5 times longer than CH9 (3.4 in paratypes); CH11 1.7 times longer than CH10 (1.7 in paratypes); CH12 2.2 times longer than CH11 (0.9 in paratypes).

Chaetigers 1 to 9 with biramous parapodia, as lateral pads (Figs. 2C, 2D, 3C, 4A, 5A); neuropodia with short, rounded to triangular, postchaetal lamella (Figs. 2C, 2D, 4A, 4B, 5A); noto- and neuropodial chaetal bundles well separated, bearing fan-shaped fascicles with numerous simple capillaries arranged in several rows (Figs. 2C, 2D); those from posterior row longer (Figs. 2C, 2D, 4B, 4C). No neuropodial acicular hooks in any anterior chaetiger. From CH10 backwards, all parapodia as elongated ridges forming almost closed flange-like cinctures near anterior margin of segment (Figs. 2E, 3A, 3J, 3K). Chaetae from CH10 arranged in two transversal rows (Figs. 4D–4H). Anterior row: CH10 with subuluncini (robust capillaries armed with long appendage) (Figs. 2G, 4D, 5B, 5C); from CH11 with subuluncini (Fig. 5C), aristate spines (Figs. 2I, 2J, 4E, 5D–5G), acicular

spines (Figs. 2K, 4F–4H, 5H), some resembling aristate spines without distal appendage (Figs. 4F–4H), and thick, slightly curved acicular spines (Figs. 2H, 4H, 5J, 5K). Posterior row with thin capillaries (Figs. 2F, 4D–4G) and robust capillaries flattened in distal half (Figs. 2F, 4E–4G, 5I), both with long distal tips. Posterior region unknown.

**Methyl Green staining.** Body stains uniformly, without any color pattern. The specimens with six or four branchial pairs do not exhibit either any methyl green staining pattern.

**Variations.** Specimens having four branchial pairs (Fig. 6A) were smaller (see morphological analyses section), but with conical prostomium anteriorly rounded, with posterolateral nuchal organs (Figs. 6B, 6C), and peristomium with two rings separated by deep dorsolateral grooves, interrupted dorsally by a dorsal prostomial crest (Fig. 6C). The prostomium and peristomium shapes were like those observed in individuals bearing more branchiae. The CH9 was also the first elongated (Fig. 6E), though in average, it was clearly shorter (0.2 mm) than in specimens with 5–6–7 branchial pairs (0.4 mm) or with 8 pairs (0.6 mm). From CH10, parapodia progressively more elongated: CH10 being 5.6 times longer than CH9; CH11 1.6 times longer than CH10 and CH12 1.6 times longer than CH11, with their chaetae arranged in two rows, forming nearly closed cinctures (Figs. 6E, 6F). However, the distribution of chaetal types along these elongate chaetigers exhibited some differences with specimens with more branchiae: the CH10 only had thin capillaries in the posterior row and slightly thicker in the anterior row (Figs. 6G–6H). From CH11, posterior row with thin and robust capillaries (Figs. 6I, 6J, 6K), and anterior row with subuluncini (Fig. 6J), aristate spines and acicular spines with deciduous distal ends (Figs. 6J–6L).

On the other hand, specimens with 5–6–7 branchial pairs were longer (Figs. 7A, 7B, 7D), with CH9 as the first elongated segment (Figs. 7A, 7E). From CH10, parapodia gradually more elongated: CH10 3.9 times longer than CH9; CH11 1.7 times longer than CH10 and CH12 1.0 times longer than CH11, with chaetae arranged in two rows, forming nearly closed cinctures. Chaetae of middle region with following chaetal patterns: CH10 with thin and robust capillaries in posterior row, and subuluncini in anterior row, some with no distal end. From CH11 (Fig. 7F), anterior row with subuluncini and aristate spines, with several lacking a distal appendage (Figs. 7G, 7J), acicular spines (Fig. 7K) and with thick acicular spines, some curved (Fig. 7L).

**Remarks.** Considering the relative size of the anterior chaetigers, *Heterospio variabilis* sp. nov. belongs to the large group of 18 longosomatid species (75% of the valid species) having eight short anterior chaetigers (CH1–CH8) and with chaetiger 9 as the first elongate segment (see *Blake & Maciolek, 2023*). Eight of these species have the chaetiger 9 longer, at least as long as the length of the first 1–4 chaetigers or as the length of the chaetigers 6–8 together. However, the other ten species have chaetiger 9 only two to three times longer than chaetiger 8: *Heterospio variabilis* sp. nov., *H. indica* (*Parapar et al., 2016*), *H. peruana* (*Borowski, 1994*), *H. africana*, *H. brunei*, *H. ehlersi*, *H. guiana*, *H. hartmanae*, *H. knoxi* and *H. paulolanai* described by *Blake & Maciolek (2023)* (Table 2).

In *H. variabilis* sp. nov. the number of branchiae is largely variable: most of the specimens had eight (16 ind.), seven (13 ind.) or six (19 ind.) branchial pairs, and less frequently four (five ind.) or five (three ind.) pairs. Nevertheless, the species can be separated from other longosomatids with CH9 only two to three times longer than CH8, due to the shape and

**Table 2 Summary of the morphological characters of *Heterospio* species without acicular hooks on neuropodia 1 and with chaetiger 9 (first elongated) only 2–3x longer than chaetiger 8.** Completed from *Blake & Maciolek (2023)*.

| Morphological characters/ Species | *H. africana* *Blake & Maciolek, 2023* | *H. brunei* *Blake & Maciolek, 2023* | *H. ehlersi* *Blake & Maciolek, 2023* | *H. guiana* *Blake & Maciolek, 2023* | *H. hartmanae* *Blake & Maciolek, 2023* |
|---|---|---|---|---|---|
| Prostomium | Conical, rounded anteriorly | Conical, tapering anteriorly | Triangular, tapering anteriorly | Triangular, tapering anteriorly | Pear-shaped, tapering anteriorly |
| Peristomium | 2 rings; large dorsal crest | 2 rings; incomplete dorsally | 1 ring; incomplete dorsally | 2 rings | 2 rings |
| Oral morphology | Narrow transverse slit; 4 lobes on anterior border, simple posterior lip; proboscis not observed | Mouth a wide transverse opening; short lobes on anterior and posterior lips | Mouth a transverse opening: a row of short lobes on posterior lip | Simple opening between 6–8 lateral lobes; proboscis sac-like | Transverse opening; 7–8 short lobes on anterior lip; proboscis short rounded sac |
| First elongated chaetiger | 9; ± 2.5x longer than CH8 | 9; ± 2x longer than CH8 | 9; ± 2.5x longer than CH8 | 9; ± 3x longer than CH8 | 9; ± 2.5x longer than CH8 |
| Chaetigers with branchiae | CH2-CH6 (5 pairs) | CH2-CH5 (4 pairs) | CH2-CH4 (3 pairs) | CH2-CH7–8 (6–7 pairs) | CH2-CH5 (4 pairs) |
| Neuropodial postchaetal lobes | Short on CH1-CH6 | Absent | Absent | Absent | Absent |
| Modified chaetae of elongated segments | Mostly encircling body from CH10. CH10-CH11: capillaries; CH12-CH13: subuluncini; from CH14 with acicular spines, rarely aristate spines | Forming cinctures from CH10. CH10: acicular spines; CH10-CH19: acicular spines and capillaries | Cinctures not present until chaetigers 20–23. CH10: acicular spines; CH11-CH23: acicular spines and capillaries | Entirely encircling body from CH10. CH10: capillaries; CH11-CH12: aristate spines and capillaries | Forming cinctures from CH10. CH10-CH24: acicula spines and capillaries |
| Posterior end | Bulbous posterior (4 chaetigers) end with curved hooks; large folds surrounding anus | Bulbous posterior end (4 chaetigers) with 1–2 acicular spines each | Bulbous posterior end (3 chaetigers) with 1–2 spines | Unknown | Bulbous posterior end (3 chaetigers) with 2 spines in each ramus |
| Depth | 55 m | 1,400–1,922 m | 60–70 m | 520–550 m | 2,470–4,950 m |
| Habitat | Sand and mud sediment | Silty clay sediment with few grain size particles; 0.9–3.5% organic carbon | —— | —— | Silty clay sediment |
| Distribution | Off Mozambique, Eastern Africa | Off Borneo, Southern China | Gulf of Thailand, Southern China | Suriname, NE South America | NW Atlantic; Off Eastern North America |
| Source | *Blake & Maciolek, 2023* | *Blake & Maciolek, 2023* | *Blake & Maciolek, 2023* | *Blake & Maciolek, 2023* | *Blake & Maciolek, 2023* |

| Morphological characters/ Species | *H. indica* *Parapar et al., 2016* | *H. knoxi* *Blake & Maciolek, 2023* | *H. paulolanai* *Blake & Maciolek, 2023* | *H. peruana* *Borowski, 1994* | *H. variabilis* sp. nov |
|---|---|---|---|---|---|
| Prostomium | Triangular, rounded anteriorly | Triangular, rounded anteriorly | Pear-shaped, tapering to rounded anteriorly | Conical, rounded anteriorly | Conical, rounded anteriorly |
| Peristomium | 1 ring, with dorsal crest | 2rings, with dorsal crest | 2 rings, incomplete dorsally | 2 rings, incomplete dorsally | 2 rings, incomplete dorsally, dorsal crest |
| Oral morphology | Two large lateral lobes and a single posterior lobe or lip | Narrow vertical slit; 2 large lateral lobes; posterior lip smooth | Mouth a transverse slit opening between 2 large lateral lobes; simple anterior and posterior lips | Unknown; proboscis sac-like | Simple opening between 2 large lateral lobes; proboscis not observed |
| First elongated chaetiger | 9; ± 2x longer than CH8 | 9; ± 3x longer than CH8 | 9; ± 2.5x longer than CH8 | 9; ± 2–3.5x longer than CH8 | 9; ± 2x longer than CH8 |
| Chaetigers with branchiae | CH2-CH8–9 (7–8 pairs) | CH2-CH8–9 (8 pairs) | CH2-CH8 (7 pairs) | CH2 to CH5 (CH6) | CH2-CH6–9 (4–8 pairs) |
| Neuropodial postchaetal lobes | Prominent on CH1-CH9 | Low flanges on CH1-CH9 | Low ridges | Absent | Short on CH4–5 to CH9 |

**Table 2** (*continued*)

| Morphological characters/ Species | *H. indica* <br> *Parapar et al., 2016* | *H. knoxi* <br> *Blake & Maciolek, 2023* | *H. paulolanai* <br> *Blake & Maciolek, 2023* | *H. peruana* <br> *Borowski, 1994* | *H. variabilis* sp. nov |
|---|---|---|---|---|---|
| Modified chaetae of elongated segments | Forming cinctures. CH10: thin and robust capillaries, highly flattened in distal half; CH11 CH13: capillaries and subuluncini | Mostly encircling body. CH10: thin and thicker capillaries; CH11: subuluncini and acicular spines; CH12-CH13: aristate and acicular spines | Nearly surrounding body. CH10: capillaries; CH11: aristate spines and capillaries; CH12: acicular spines and subuluncini | Forming cinctures. CH10: subuluncini and capillaries; CH11-CH12: thickened, aristate spines, many lacking aristae, acicular spines and capillaries | Forming cinctures. CH10: subuluncini; CH11-CH13: aristate and acicular spines; from CH14 with thick acicular spines |
| Posterior end | Bulbous (5 chaetigers) with 2–4 acicular hooks each | Unknown | Unknown | Bulbous (5 chaetigers) with recurved hooks | Unknown |
| Depth | 2.5–22 m | 13–61 m | 69 m | 4,125–4,423 m | 7–106 m |
| Habitat | Mostly clayey silt and sandy silt sediments | Grey sands to fine muddy sands; 13–13.3 °C | —— | Manganese nodule area | 49–94% fine sand; 13.2–30 °C; 34.2–36.06 ups; 2.4–8.4% organic matter; 1.03–5.4 ml/L oxygen |
| Distribution | Malvan, Western India; Sudan, Arabian Sea | North Island, Hawke Bay, New Zealand | Off NE Brazil | Peru Basin; Peru-Chile Trench | Gulf of California |
| Source | *Parapar et al., 2016*; *Blake & Maciolek, 2023* | *Knox, 1960*; *Blake & Maciolek, 2023* | *Blake & Maciolek, 2023* | *Borowski, 1994*; *Blake & Maciolek, 2023* | This study |

distribution of chaetae in its elongated segments, the shape of the neuropodial postchaetal lamellae or the form of the prostomium and peristomium. For example, among the other species with eight branchial pairs (1) in *H. indica* the prostomium is triangular, the peristomium has one ring, prominent neuropodial postchaetal lamellae, thin and thicker capillaries in CH10 and subuluncini and thin capillaries from CH11; (2) *H. knoxi* has a triangular prostomium, with a dorsal crest extending to chaetiger 2, rounded neuropodial postchaetal lamellae as low flanges, CH10 bearing thin and thicker capillaries, and from CH11 with subuluncini, aristate chaetae and acicular spines (*Blake & Maciolek, 2023*) (Table 2). On the contrary, *H. variabilis* sp. nov. has a conical prostomium, tapering to a rounded tip, with neuropodial postchaetal lamellae short and rounded and, from CH10, a large chaetal variety: subuluncini, aristate spines, blunt acicular spines and thick, curved acicular spines.

*Heterospio variabilis* sp. nov. can also be separated from those species having 6 or 7 branchial pairs: *H. guiana* which has a triangular prostomium, tapering to a narrow tip, and only bears capillaries in CH10 and aristate spines in CH11-CH12 (Table 2). On the other hand, *H. paulolanai* has a pear-shaped prostomium with a narrow-rounded tip, neuropodial postchaetal lamellae as low ridges and bears capillaries in CH10, and from CH11, aristate spines, acicular spines and subuluncini. This last species was described from only one individual with branchiae on chaetigers 2–8 (seven pairs), but due to damage from an earlier dissection, the presence of branchiae on CH9 remains in doubt (*Blake & Maciolek, 2023*).

*Heterospio variabilis* sp. nov. also showed important morphological differences with those species having few branchiae: *Heterospio africana*, with five branchial pairs, has a wide and large dorsal crest on the peristomium, the neuropodial postchaetal lamellae are short, bear only capillaries in CH10 and CH11, and capillaries and subuluncini in CH12-CH13

(Table 2). Furthermore, the two species bearing four branchial pairs lack neuropodial postchaetal lamellae and the chaetae in elongated segments are distinct: *H. brunei* only bears acicular spines in CH10-CH19, while *H. hartmanae*, with a pear-shaped prostomium, has acicular spines in CH10, and capillaries and acicular spines in CH11-CH24 (Table 2).

*Heterospio peruana* was described as bearing four branchial pairs, but two individuals bearing five pairs and small specimens with one to three pairs were also reported (*Borowski, 1994*). This longosomatid has also a wide variety of chaetae on elongated segments: capillaries, subuluncini, aristate chaetae and acicular spines from chaetiger 10, although some paratypes lack subuluncini and aristate chaetae (*Borowski, 1994*).

Thus, besides the fact that most specimens of *H. variabilis* sp. nov. have six to eight pairs and only five individuals with four pairs and three individuals with five pairs were found, it can be also separated from *H. peruana* by the presence of robust capillaries flattened in their distal half in the posterior row, and because thick curved acicular spines are also present in the anterior row of some elongated segments (Table 2).

Before the *Blake & Maciolek (2023)* study, besides the everted sac-like proboscis on some specimens, the mouth features had not been examined in any description of a longosomatid species. They examined the anterior ventral region of 23 species and found a wide range of morphological characteristics, so that they suggested that further research will fully demonstrate that the oral morphology is a useful taxonomic character. In this description, we observed that the mouth of *H. variabilis* sp. nov., is a simple opening between two large lateral lobes, different from the oral morphology observed in close species (Table 2). The mouth of those species with only two large lateral lobes, displays other additional structures: *H. indica* has a single posterior lip, *H. knoxi* also exhibits a posterior lip formed by the anterior edge of the peristomium, and *H. paulolanai* additionally bears simple anterior and posterior lips (Table 2).

**Etymology.** The term "*variabilis*" to name the new species was chosen to emphasize its high variability in the number of branchiae.

**Habitat.** At 7 to 106 m depth, in sediments with 49–94% of fine sand, temperatures of 13.2–30 °C, 34.2–36.06 psu of salinity, 2.4–8.4% of organic matter and 1.03–5.4 ml/L of dissolved oxygen.

**Distribution.** Widely recorded in the eastern continental shelf of the Gulf of California.

## Morphometric analyses

The ninth chaetiger was the first elongated segment, and is an invariant character in all examined individuals. The length and variability of the elongated segments gradually increased towards the posterior region: ch9L (mean = 0.41 mm; CV = 40.95), ch10L (mean = 1.62 mm; CV = 44.17), ch11L (mean = 2.72 mm; CV = 51.60) and ch12L (mean = 2.95 mm; CV = 55.84) (Table S1). The rate of length Rch9/anL (mean = 0.13; CV = 38.95), the total length (mean = 10.91 mm; CV = 32.35), width of anterior region (mean = 0.54 mm; CV = 30.39) and the prostomium length (mean = 0.28 mm; CV = 27.66) were less variable. On the contrary, the prostomium width (mean = 0.32 mm; CV = 24.3), length of CH1 to CH8 (mean = 2.92 mm; CV = 24.16) and the number of branchiae (mean = 6.57 mm; CV = 18.55) presented the lowest variability (Table S1).

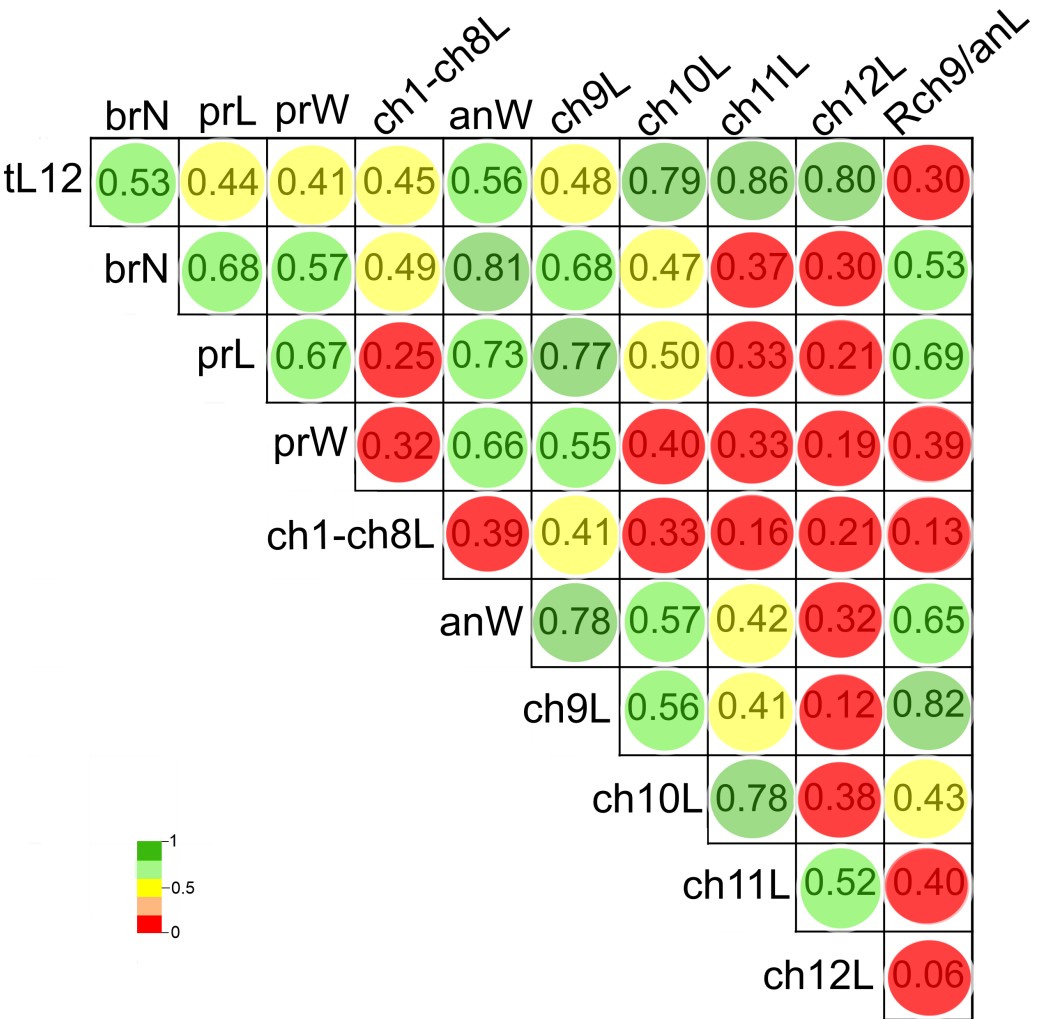

**Figure 8** **Pearson's correlation matrix between the total length and the 10 other examined morphological characters of *Heterospio variabilis* sp. nov.** Abbreviations: brN, number of branchiae; prL, length of prostomium; prW, width of prostomium; ch1-ch8L, length of chaetiger 1 to chaetiger 8; anW, width of body; ch9L, ch10L, ch11L, ch12L, length of chaetiger 9, 10, 11, 12, respectively; Rch9/anL, rate between length of chaetiger 9 and length of anterior region.

The length of chaetigers 10 ($r = 0.79$), 11 ($r = 0.86$) and 12 ($r = 0.8$), the width of anterior region ($r = 0.56$), number of branchiae ($r = 0.53$) and length of chaetiger 9 ($r = 0.48$) exhibited the higher Pearson correlation with the total length of specimens (tL12). In general, the larger individuals were wider, with chaetigers 9 to 12 longer and with more branchial pairs. On the contrary, the length and width of the prostomium, the length of CH1-CH8 and the rate of length Rch9/anL exhibited the lowest correlation with the specimen's length ($r < 0.45$). In particular, the number of branchiae was highly correlated to the width of the anterior region ($r = 0.81$), length of chaetiger 9 ($r = 0.68$), length of prostomium ($r = 0.68$), width of prostomium ($r = 0.57$), and the rate of length Rch9/anL ($r = 0.53$) (Fig. 8, Table S2).

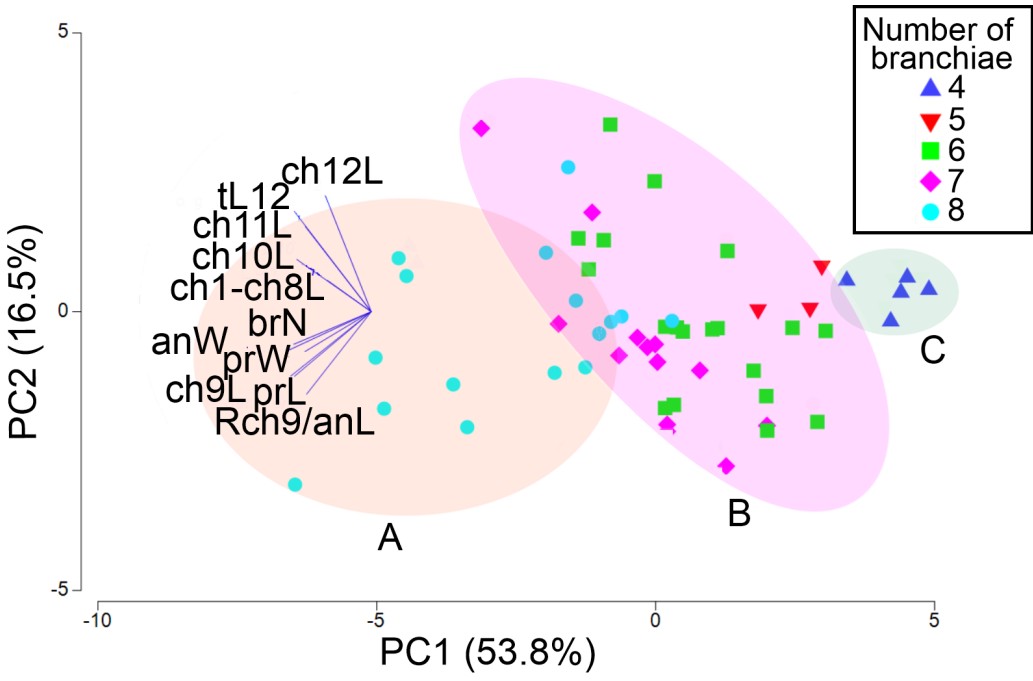

**Figure 9** **PCA based on 11 morphological characters; specimens labeled according to their number of branchial pairs.** Shadow A = eight branchial pairs; B = five-six-seven pairs; C = four pairs. Abbreviations: brN, number of branchiae; prL, length of prostomium; prW, width of prostomium; ch1-ch8L, length of chaetiger 1 to chaetiger 8; anW, width of body; ch9L, ch10L, ch11L, ch12L, length of chaetiger 9, 10, 11, 12, respectively; Rch9/anL, rate between length of chaetiger 9 and length of anterior region.

The first two PCA components explained 70.3% of the total morphological variation (Fig. 9). The axis 1 of PCA described the highest variance of the model (53.8%), with the most important explanatory variables being the width of the anterior region (−0.36) and the length of chaetiger 9 (−0.35); the axis 2 of PCA only accounted for 16.5% of variability, mainly linked to the length of chaetiger 12 (0.5) and the total length (0.43). The number of branchiae has been typically used as a diagnostic character to differentiate the species, but in this case, it only contributed with −0.33 (PC1) and −0.14 (PC2) to explain the total variation (Table S3).

The addition of specimens on the PCA plot, arranged according to their number of branchial pairs because, among the characters correlated with the body size, it presented the lowest variability (CV = 18.55), suggested the presence of different morphological groups. Significant differences were found between individuals with eight or four branchial pairs ($R_{ANOSIM} = 0.83, p = 0.001$) and between specimens with eight or six pairs ($R_{ANOSIM} = 0.41$, $p = 0.001$), but the individuals with five or six branchial pairs ($R_{ANOSIM} = 0.05, p = 0.583$) and those with six or seven pairs ($R_{ANOSIM} = 0.04, p = 0.218$) integrated the same group (Fig. 9).

As a result, the individuals were classified in three morphological groups, distinguished by the presence of eight (morphotype A), five-six-seven (morphotype B) or four (morphotype C) branchial pairs (Fig. 9), whose differences were tested by a discriminant analysis. The

Wilks' lambda value of 0.093 was highly significant ($F_{(12,96)} = 18.2$, $p = 0.0001$), supporting the hypothesis that the examined individuals could be assembled in three morphological groups; the analysis also showed that the individuals were appropriately classified inside the corresponding group. The forward stepwise way removed five morphological variables from the discriminant function model (F*value* < 1), so, six variables remained to differentiate the groups (Table S4). Subsequently, the partial Wilks' lambda selected the number of branchiae (brN) (0.57) as the most important variable to the discriminant function, followed by the width of the anterior body (anW), the width of the prostomium (prW), the rate length Rch9/anL, the length CH1–CH8 (ch1–ch8L) and the length of CH9 (Table S4). The first three variables exhibited tolerance values larger than 0.5 but, as initially reported, except for the prostomium width, they were highly correlated with the total length of specimens.

The plot of the discriminant functions confirmed the separation of the three suggested morphotypes, with the first canonical root explaining 94.8% of the variance (Fig. 10), which was mainly defined by variables associated with the body length and number of branchiae (Table S5). It clearly separated the specimens with 8 branchial pairs having the longer CH1–CH8 (ch1–ch8L) and CH9 and higher rate Rch9/anL, from those specimens with only 4 branchial pairs with shorter ch1-ch8L, ch9L and r Rch9/anL; the individuals with 5–6–7 branchial pairs presented intermediate values in these characters. The second canonical root only explained 5.2% of variance and thus, was barely relevant to discriminate the morphotypes, but exhibited the great variability of each group (Fig. 10). The multivariate analysis showed that the number of branchiae and length of some chaetigers determined the presence of three morphological groups. However, their great variability and high correlation with the body length, together with the few differences found in the morphology of specimens with different number of branchiae, clearly showed that they could not be considered as separate taxa. In particular, the number of branchiae has been regularly used to separate taxonomically the longosomatids but, until the importance of this character and other diagnostic characters to discriminate species are fully understood, as well as their variability with specimens' size, we cannot conclude that the examined specimens here belong to distinct species.

## Distribution patterns

*Heterospio variabilis* sp. nov. was found along the eastern shelf of the Gulf of California, mostly in the winter-spring season (46 ind.), whereas in the summer-autumn only 9 individuals were collected. During the winter-spring, the highest abundances were found in Rocas Consag (Sta. 37 = 30 ind.; Stat. 38 = 2 ind.) in the northern gulf, and El Fuerte River (Stat. 50 = 10 ind.; Sta. 52 = 2 ind.; Stat. 51 = 1 ind.) in the south (Table 1, Fig. 1). In the summer-autumn, it was also collected in both localities, but clearly with lower abundance (Sta. 37 = 3 ind.; Sta. 50 = 1 ind.; Sta. 52 = 1 ind.). In the far southern gulf, during the winter-spring season, only one individual in Mazatlán Bay (Sta. C8–7) and another in Punta Mita (Sta. 61) were found.

Morphotype B (five-six-seven branchial pairs) was the most abundant (35 ind.), followed by morphotype A (eight pairs) with 16 specimens. The higher abundances of both

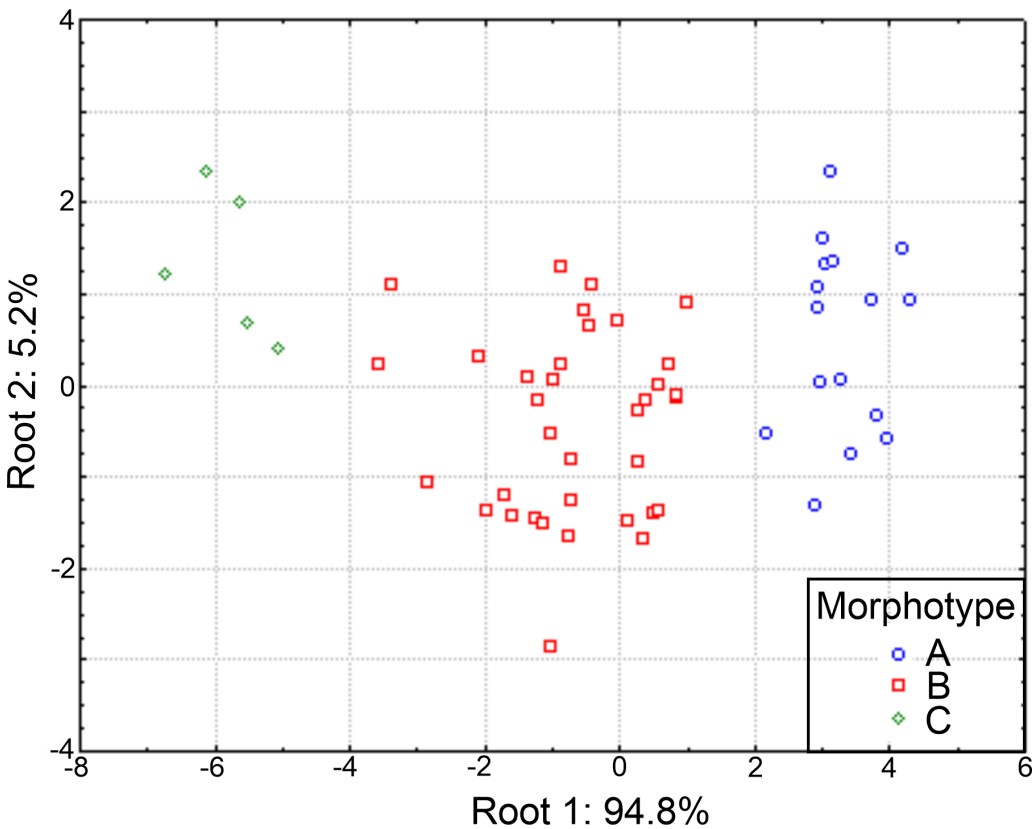

**Figure 10  Canonical analysis based on the first and second discriminant functions.** Abbreviations: A, morphotype A (4 branchial pairs); B, morphotype B (5–6–7 pairs); C, morphotype C (8 pairs).

morphotypes were found in the same localities: Consag Rocks and in front of El Fuerte River. On the other hand, the five specimens of morphotype C (4 branchial pairs) were exclusively collected in front of El Fuerte River (Table 1, Fig. 1).

The axis 1 of PCA explained 61.8% of the environmental variability where the new species was distributed, basically associated with dissolved oxygen (−0.49) and depth (0.484) changes. The axis 2 of PCA accounted for 20.2% of variance, mainly related to temperature fluctuations (0.872) (Table S6). The highest abundance of morphotypes B and A was found at 30.3 m depth in the northern gulf, on well oxygenated bottoms (5.4 ml/L) and with low organic matter percentage (2.4%) (Fig. 11). However, these morphotypes were basically collected in the same sampling stations and no environmental differences were found where they dwell. On the contrary, morphotype C was only collected in front of El Fuerte River, in the central gulf, at 97 m depth, with lower oxygen levels (1.47 ml/L) and higher organic matter concentrations (5.7%). Seasonally, the higher abundances were found in the winter-spring, where the lower temperatures were recorded (mean = 15.35 °C against 24.94 °C in summer-autumn) (Table 1).

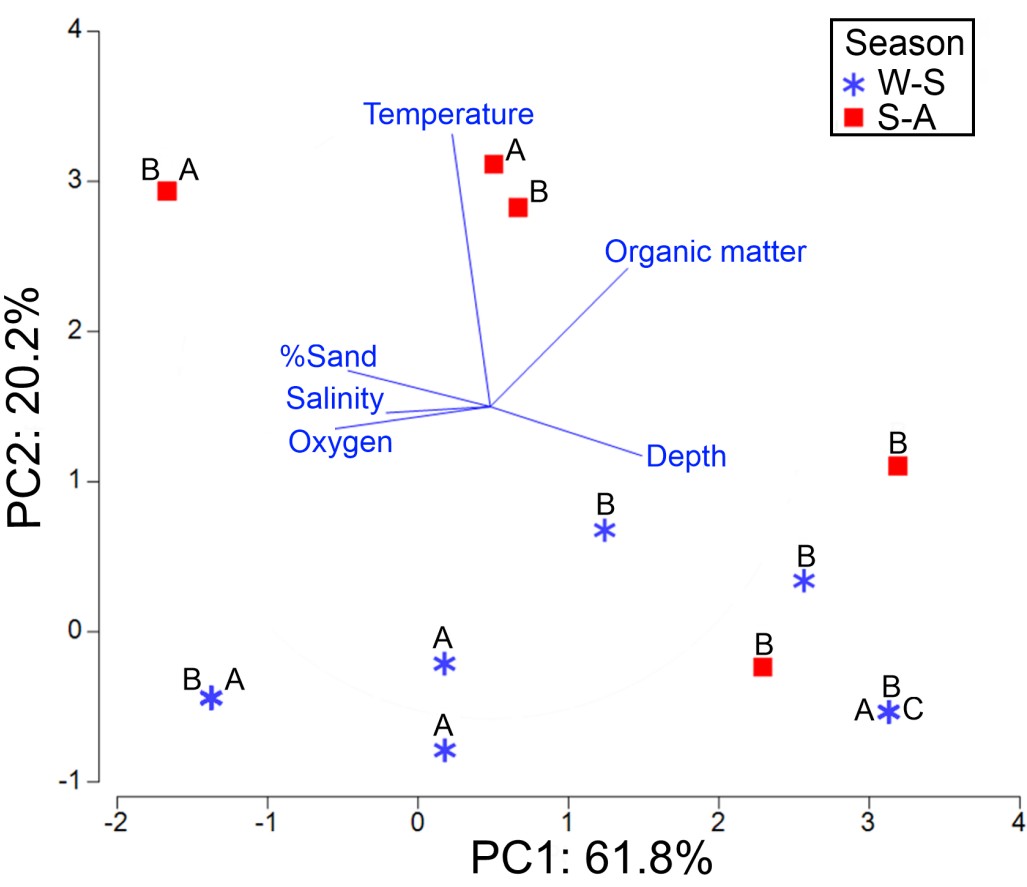

**Figure 11** **PCA for the first two components based on the environmental conditions where the three morphotypes (A, B, C) were found.** Stations were labeled according to their sampling season: W-S, Winter-Spring season; S-A, Summer-Autumn season.

## DISCUSSION

### Taxonomy and morphology

Of the 23 species of the family Longosomatidae currently recognized, 13 of them (56.5%) were recently described by *Blake & Maciolek (2023)*. Following this important publication, the taxonomy and the importance of the diagnostic characters in this family could be better understood, especially when more intraspecific morphological variability was observed in several taxa. However, the scarce knowledge about the taxonomy of longosomatids can be perceived, for example, when the term "palps" is used to designate the grooved feeding structures (*Blake & Maciolek, 2019*). According to *Blake & Maciolek (2023)* they should be named "dorsal tentacles", since there are evidences that longosomatids are more closely related to cirratulids than to spionids (*Rouse, Pleijel & Tilic, 2022*). However, we decided to use here the term palp until those phylogenetic relationships are fully elucidated.

Indeed, the presence of dorsal palps has been described and/or drawn for several species (*Hartman, 1965*; *Wu & Chen, 1966*; *Laubier, Picard & Ramos, 1972-1973*; *Borowski, 1994*; *Parapar et al., 2016*; *Blake & Maciolek, 2023*). However, they are easily lost during

the collecting and fixation processes and are usually missing in the specimens examined (*Parapar, Aguirrezabalaga & Moreira, 2014*). This has caused uncertainties about the actual presence of their dorsal palps, and some authors as *Uebelacker (1984)*, *Parapar, Aguirrezabalaga & Moreira (2014)* and *Bochert & Zettler (2009)* have interpreted the deep grooves behind the prostomium as nuchal organs and not dorsal palps' scars (*Parapar et al., 2016*). *Parapar, Aguirrezabalaga & Moreira (2014)*, for example, in their description of *H. reducta*, indicated that the unique structure observed under the SEM, was the deep groove between the prostomium and peristomium, which did not look like a palp or scar, but rather like a nuchal organ. Notwithstanding, *Parapar et al. (2016)* in their description of *H. indica* and *Blake & Maciolek (2023)* for *H bathyala*, *H. catalinensis* and *H. hartmanae*, confirmed the presence of dorsal palps. Here we also confirm the presence of a dorsal palp in a SEM photo for *H. variabilis* sp. nov. (Figs. 3A–3E). In a paratype (CNAP-ICML: MEB-POP–13–001), a large dorsal palp is attached on the right side of the peristomium but, in addition, posterolateral to the prostomium, the nuchal organs, as deep grooves with cilia, are also shown.

Although the presence of distinct types of modified chaetae and their distribution along the elongated segments in the examined specimens did not exhibit any clear trends, some signals indicated that, as *Borowski (1994)* and *Parapar, Aguirrezabalaga & Moreira (2014)* previously suggested, they could be transitional stages from capillaries to acicular spines, associated with the developmental state of individuals. Therefore, it is necessary to examine in detail complete specimens with different sizes and to compare their variations with other longosomatid species to detect some eventual pattern and then validate its importance as a diagnostic character.

## Morphometry

The identification of longosomatid species has been traditionally based on the number of branchial pairs, the number of short anterior segments, the location of the first elongated segment and its length relative to preceding and following segments, the presence of modified neurochaetae on chaetiger 1 and the chaetal types in abdominal segments (*Wu & Chen, 1966*; *Borowski, 1994*; *Parapar, Aguirrezabalaga & Moreira, 2014*; *Parapar et al., 2016*; *Blake & Maciolek, 2023*). Therefore, the ample variability recorded in the examined individuals of this study in some of their diagnostic characters, as the number of branchiae or the relative length of the first elongated segment, for example, could be interpreted as meaning that they belong to different species. However, some previous species descriptions have shown that their morphological variations could be related to their body size (*Borowski, 1994*; *Parapar et al., 2016*).

Although the morphological intraspecific variations in relation to body size have been poorly explored in longosomatids, *Borowski (1994)* observed a possible association between the number of branchiae and the individual size. He found that small specimens of *H. peruana* have one to three branchial pairs and that with the increase of the specimens' size, they reach four pairs, but the larger specimens do not necessarily bear more branchiae. These likely relationships with body size were also observed in other species, such as *H.*

*indica*, whose large specimens bear seven to eight branchial pairs, while those smaller have four pairs (*Parapar et al., 2016*).

In *H. variabilis* sp. nov., both large and small specimens have different numbers of branchiae, but the material examined provided new insights to recognize the relationships between the morphological variation and the body size in longosomatids. The examination of 11 characters by multivariate analyses showed that the observed differences between the suggested morphotypes are precisely related to the specimen length. In fact, the presence of three morphological groups, considering the number of branchiae, was confirmed by the discriminant analysis, but the main characters explaining their separation: number of branchial pairs and length of the first elongated segment (CH9), were related to the individual's length. However, these characters presented many variations, so that they were not entirely size-correlated: for example, in smaller specimens (<eight mm), between four and seven branchial pairs are present, but larger individuals (>12 mm) can bear between 6 and eight pairs; also, chaetiger 9 was from 0.13 to 0.53 mm long in small individuals, but from 0.33 to 0.97 mm in larger individuals. Therefore, we propose that the three detected groups are morphotypes of a single species, *H. variabilis* sp. nov., whose morphological variability is largely dependent on the body size. Unfortunately, no gametes were seen in the collected specimens to consider whether the morphotypes correspond to different ontogenetic stages, since among the longosomatids, the morphological variations associated with ontogeny or development have not been studied yet.

Therefore, it would be necessary to examine more specimens, particularly short individuals bearing few branchial pairs, to corroborate the variations or homogeneity of the analyzed characters. In addition, due to dependence of several morphological characters to the individual size, in future descriptions of species, it will be necessary to analyze whether other morphological characters, such as length of the anterior region (ch1-ch8L), length of the first elongated segment (ch9L) or the rate of length Rch9/anL, could be also appropriate to differentiate the species in this family.

## Ecology

The publication of new records of longosomatid species are very important, since their reports are very scarce, and the localities where they were collected are scattered around the world seas (*Borowski, 1994*; *Blake & Maciolek, 2023*). Although the interpretations about their distribution patterns are difficult to establish, six of the seven species having eight short anterior chaetigers and chaetiger 9 as the first elongate segment, have only been recorded in the Pacific Ocean. Among them, three species are so far exclusively distributed in the Eastern Pacific: *H. catalinensis* from southern California and the Gulf of California, in shelf and deep-sea habitats (*Hartman, 1944*; *Mendez, 2006*; *Mendez, 2007*), *H. peruana* from abyssal depths of Peru Basin (Borowsky, 1994) and now *H. variabilis* sp. nov. from the Gulf of California shelf (Table 2). It is unlikely that the records of *H. catalinensis* from the Gulf of California at 1,550–1,590 m by *Mendez (2006)* and *Mendez (2007)* belong to this species because of their habitat, at a much greater depth (*Blake & Maciolek, 2023*).

The longosomatid species described so far, have been recorded in marine regions too distant between them and in large depth intervals, but mainly in the continental slope and

abyssal depths (*Blake & Maciolek, 2023*). Although they were mainly found in soft bottoms, and being potentially subsurface deposit feeders (*Jumars, Dorgan & Lindsey, 2015*), little is known about their biology and ecology (*Blake & Maciolek, 2023*). The distribution of *H. variabilis* sp. nov. is, so far, limited to the continental shelf, but it lives on a large range of environmental conditions. Its highest abundances were recorded in the winter-spring season, linked to a decrease in temperature and organic matter percentage. However, no relationships were found between the environmental conditions and the occurrence of the three morphotypes found in the Gulf of California. The morphotypes B (five-six-seven branchial pairs) and A (eight pairs) were the most abundant, but they were practically collected in the same sampling stations, and no environmental preferences were detected. However, morphotype C (four branchial pairs) was only found in the outer shelf, subjected to low oxygen levels and higher organic matter percentages. Despite the fact that extensive benthic sampling on large areas of the Eastern Pacific Ocean, the Central Pacific gyre, and Antarctica have been carried out and no specimens of *Heterospio* were found (*Blake & Maciolek, 2023*), it is necessary to get more information about the habitats where the species occur, to understand the effect of environmental factors on the settlement and development of the longosomatids.

## CONCLUSIONS

We can conclude that *Heterospio variabilis*, the new species described here, does differ significantly from other close species of longosomatids. The morphological variability found in the individuals examined is remarkably high, although not sufficient to separate them into different species. Several of the 11 characters analyzed by multivariate techniques often overlap between the specimens with different number of branchiae, a character usually considered the strongest character to separate species in this family. The distribution and environmental conditions measured, where the morphotypes with four, five-six-seven or eight branchial pairs were found, does not allow us to determine any pattern that would help to separate them either in different species. Therefore, we believe that the new species displays a high morphological plasticity, and although such variability has seldom been found in other longosomatid species, until more studies are carried out with more class sizes to examine, we think that all individuals belong to only one species, *H. variabilis* sp. nov.

## ACKNOWLEDGEMENTS

We would like to thank M Hendrickx, head of the institutional project ''Cortes'', for inviting both of us to participate in the oceanographic expeditions. We especially thank Alexandra Rizzo and two anonymous reviewers for their valuable comments, which considerably improved the manuscript.

### Funding

The work was financially supported by the Instituto de Ciencias del Mar y Limnología, ICML- UNAM, Mexico. The funders had no role in study design, data collection and analysis, decision to publish, or preparation of the manuscript.

### Grant Disclosures

The following grant information was disclosed by the authors:
The Instituto de Ciencias del Mar y Limnología, ICML- UNAM, Mexico.

### Competing Interests

The authors declare there are no competing interests.

### Author Contributions

- Pablo Hernández-Alcántara conceived and designed the experiments, performed the experiments, analyzed the data, prepared figures and/or tables, authored or reviewed drafts of the article, and approved the final draft.
- Vivianne Solis-Weiss conceived and designed the experiments, analyzed the data, authored or reviewed drafts of the article, obtained the financing for this work, and approved the final draft.

### Data Availability

The datasets are available in the Supplementary Files.

The morphometric variables are available at SEANOE: Hernández-Alcántara Pablo, Solís-Weiss Vivianne (2023). Morphometric dataset of Heterospio variabilis (Annelida, Longosomatidae) from the Gulf of California shelf, Eastern Pacific. SEANOE. https://doi.org/10.17882/96906.

### New Species Registration

The following information was supplied regarding the registration of a newly described species:

Publication LSID: urn:lsid:zoobank.org:pub:F3462F09-2330-42F3-BA76-C2ACDEE10504

*Heterospio variabilis* Hernández-Alcántara & Solís-Weiss sp. nov. LSID: urn:lsid:zoobank.org:act:C8DF52A4-B1F1-4F9B-83C4-5A410414A9E7.

### Supplemental Information

Supplemental information for this article can be found online at http://dx.doi.org/10.7717/peerj.17093#supplemental-information.

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
