# Peer review of "Morphometric and taxonomic approach to describe Heterospio variabilis (Annelida, Longosomatidae), a new species with three size-dependent morphotypes, from the Gulf of California, Eastern Pacific"

_PeerJ, doi:10.7717/peerj.17093_

## Round 0.1 · original submission · Major Revisions

Many thanks for your submission to PeerJ. I have received three reviews that indicated that the manuscript is overall well written and with valuable results but still need some additional work. Please see attached the reviews and the commented manuscript file. There is a recent book chapter with new terminology for the family including descriptions of oral morphology and I suggest the authors to follow it or justify why this recent publication is not being used (see Blake & Maciolek 2023). In relation to Figure 9 and 10, add labels for the abbreviations in the figure legend.

Reviewer 1 ·

Basic reporting

1. Clear and unambiguous, professional English used throughout.
Overall, the paper is well written. However, there are areas where the English needs some editing and the sentences revised for clarity. I have done this in part in the attached file.

2. Literature references Sufficient.
The references cited are relevant and necessary to support the text.

3. Professional article structure, figures, tables.
The article is organized as expected for the Professional article. For the most part, the figures are well done and necessary.

Figure 6, however, which is used to illustrate Methyl Green staining patterns should be deleted. There are no patterns to the strain, only solidly green worms.

None of the figures: 2, 3, 4, 5, 6, 7 or 8 have any indication as to which specimen was illustrated. Each figure caption should provide the referenced Museum number for each specimen illustrated. The most relevant example to follow for referencing specimens in Figure captions is Blake & Maciolek (2023).

Tables: Table 1 includes location, specimen count, and environmental data for each sample collected. However the date of sample collection should also be included. Since these samples were collected more than more than 35 years ago, the data presented is not recent.

Table 2 includes morphological comparison of the new species and 9 other species considered to be the most similar. Oral morphology was not included, also not in the basic description.

The authors have not described the oral morphology, whereas Blake & Maciolek (2023) reported and illustrated the oral morphology for all their species. There are differences that may be of importance at the species level. Was the proboscis everted on any of their specimens?

4. Self-contained relevant to the hypothesis.
In this case, the hypothesis is that there is a new species and the morphotypes are growth stages. The results support the new species and that there is only a single species.

Experimental design

1. Original primary research within Aims and Scope of the journal
The aim of this paper is to describe and document a new species of polychaete, genus Heterospio. This is within the scope of this journal based on previous publications I have seen.

2. Research question well defined, relevant & meaningful. It is stated how research fills an identified knowledge gap.
The materials examined are based on collections of an unidentified species of Heterospio. The aim of the paper is to describe it and justify it as a new species. This has been accomplished and the knowledge gap has therefore been identified and filled.

3. Rigorous investigation performed to a high technical & ethical standard.
The relevant literature has been reviewed and cited where relevant. The new species description has been presented as required to report a new species of polychaete. This includes the ZooBank references.

4. Methods described with sufficient detail & information to replicate.
The methods including the statistics used are basic and should be suitable for replication assuming the raw data is correct.

Validity of the findings

1. Impact and novelty not assessed. Meaningful replication encouraged where rationale & benefit to literature is clearly stated.
This study is most important to an audience interested in marine benthos and systematics of polychaetes, especially from the eastern north Pacific. The novelty of this study is the identification and description of a rather rare type of polychaete from the Gulf of California. It is compared with its most similar congeners from other geographic areas, none of which also occur in the Eastern Pacific in shallow waters.

2. All underlying data have been provided; they are robust, statistically sound, & controlled.
All of the specimens used in this study have been documented and their deposition has been recorded. Data used for the statistical analysis has been made available in Supplemental files. The statistics used are sound, but there is no information provided on how or if the results were independently confirmed. There is no information provided relative to the quality of the environmental data used in the ecological summary.

3. Conclusions are well stated, linked to original research question & limited to supporting results.
In general, the conclusions are well stated. There is a new species and morphological variability is determined to due to growth related size differences.

Additional comments

1. This reviewer has edited some of the text in Track Changes and provided comments in the margins. Several issues were noted.

2. The authors make reference to a previously “known species” identified in the Gulf of California, presumably Heterospio catalinensis. They imply that the specimens have been determined to not be a separate species and that all specimens from the Gulf of California were referred to Heterospio sp. 1 by Hernándes-Alcántara & Solís-Weiss (2005). However, I have examined that paper and find no reference to any other species of Heterospio, including H. catalinensis. I note, however, that Blake & Maciolek (2023: 48) stated: “Records of H. catalinensis from the Gulf of California in 1550–1590 m by Mendez (2006, 2007) are likely not this species because of their much greater collection depth.” The authors do not make any reference to the works of Mendez which is unfortunate because this is most likely another undescribed species of Heterospio from the Gulf of California, but from deep-water.

3. The map shows records from Mazatlán and Mita Point, which are not in the Gulf of California. Is it possible the new species ranges further south along the Mexican coast?

4. The authors make the following statement: “Thus, at present, the longosomatids comprise 23 valid species and four unnamed species, but it is likely that many species have still to be discovered.” I would be cautious about such a statement. Blake & Maciolek (2023) noted that in general, species of Heterospio are rare globally and I would not expect that many more species are actually out there. Blake & Maciolek (2023) had access to an unusually broad scope of collections, some that were part of large regional baseline and monitoring surveys. They also note that despite extensive benthic sampling, large areas of the Eastern Pacific Ocean, Central Pacific gyre, and Antarctica have not yielded a single specimen of Heterospio.

5. The authors need include the abbreviations in the Figure captions for figures 9 and 10. The readers should not be expected to find these in the Methods. Otherwise these figures become incomprehensible.

Annotated reviews are not available for download in order to protect the identity of reviewers who chose to remain anonymous.

Reviewer 2 ·

Basic reporting

Clear, unambiguous, professional English is used throughout. The introduction contains the necessary background information and a thorough presentation of existing literature on the topic. The structure is conform to the journal’s standards.

The text is supported by a large amount of figures, with good quality SEM photographs. MG photographs quality could be improved. Photographs of unstained specimens and specimens stained with Shirlastain A could be relevant, although the illustrations and SEM provided are certainly sufficient. The organization of the Sem figures is a bit inconsistent. The lettering and scale bars are placed rather chaotically (probably because the pictures themselves are placed so). The fonts could be more adapted to the general font size of the paper. In addition, some legends on the figures vary in upper/lower case, for example “dorsal tentacle scars is sometimes “dTs”, sometimes “dtS”. The general convention is that a legend on a figure starts with a lower case, and upper case letters are introduced to signify the beginning of a second word in that legend. For example:

pr: prostomium
dT: dorsal tentacles
dCr: dorsal crest
dTS: dorsal tentacle scar
ch: chaetiger

These are simply conventions though, the most important is that the labeling is consistent.

Most of the raw data is supplied. However, the details of individual specimens measurements is missing, only the summary is presented for each character. If possible, this should be provided.

Experimental design

The research fits the scope of the journal. The research question is well defined, relevant and meaningful. The reasons and context for this research are clear. Each analysis, morphological and morphometrics, was well performed and relevant. The methods used are well described.

Validity of the findings

The analysis performed are relevant for this research question and the results are well supported. There is no doubt that this is a new species. Comparisons with other species in the group are detailed. Intraspecific variations have been more than thoroughly explored (more than is usually done). The different hypothesis that could explain such variations are all presented, explained, and when possible, tested.

Additional comments

In addition to the three sections above, I have some minor comments on typos/language or such:

Line 80: “So, the aim of the present study…”. This is perhaps more oral language than written language. “Therefore” might be more appropriate or simply “The aim of…”.
Line 82: “assessed its interspecific”. Do you mean “intraspecific”?
Line 77 to 94: This paragraph contains a mix of present and past tenses, it might be more relevant to pick one and stick to it.
Line 264: I think there might be a mistake in the CH12/11 ratio for the paratypes? 0.9 would mean that CH12 is shorter than CH11 on average, when it is 2.2 times longer in the holotype.
Line 352: Unit for salinity should be “psu” and not “ups”
Figure 3B and E: It might be relevant to indicate that these were mirrored, as you precise in the text (I think) that one specimen had a tentacle attached on the right side. These have it on the left but seem to be of the same specimen as 3A, C-D. Even though it’s slightly confusing when looking at the figure initially, it’s easy to figure out the mirroring. But it might be worth mentioning it in the legend.
Figure 4: The black lettering is difficult to read. I would suggest using white or writing in the black space around the figure and using arrows/lines.
Figure 5E: the legend says “Chaetiger 7-9”, the image is labeled with CH7, CH9, CH10.

Figure legends in general: I would recommend repeating abbreviations in each legend. Both for the morphology and plots. It makes reading much easier than when one has to go back and forth between the figures and the methodology section. Even if the abbreviations are not particularly cryptic, figures should be understandable from their content and legend only.

In the end, the article is very good. I have nothing to say about the content, so I recommend minor revision only because of these few corrections that are mostly cosmetic.

·

Basic reporting

No comment.

Experimental design

No comment.

Validity of the findings

No comment.

Additional comments

The manuscript may be accepted after minor changes. Suggestions can be found throughout the text.

---

## Round 0.2 · accepted · Accept

Dear authors,

Many thanks for accepting the reviewer's suggestions. The reviewers and I have agreed that this manuscript is ready for publication.

Reviewer 1 ·

Basic reporting

This paper now includes relevant text that addresses earlier concerns and is now acceptable for publication.

Experimental design

This paper now includes relevant text that addresses earlier concerns and is now acceptable for publication.

Validity of the findings

This paper now includes relevant text that addresses earlier concerns and is now acceptable for publication.

Additional comments

I have no additional comments.

Reviewer 2 ·

Basic reporting

The authors addressed all the comments each reviewer made adequately and the manuscript seems now ready for publication.

Experimental design

No further comments

Validity of the findings

No further comments

Additional comments

No further comments